# Large-scale genome-wide analysis links lactic acid bacteria from food with the gut microbiome

Edoardo Pasolli[1,2], Francesca De Filippis [1,2], Italia E. Mauriello[1], Fabio Cumbo [3], Aaron M. Walsh[4,5], John Leech[4,5], Paul D. Cotter [4,5], Nicola Segata [3] & Danilo Ercolini [1,2 ✉]

Lactic acid bacteria (LAB) are fundamental in the production of fermented foods and several strains are regarded as probiotics. Large quantities of live LAB are consumed within fermented foods, but it is not yet known to what extent the LAB we ingest become members of the gut microbiome. By analysis of 9445 metagenomes from human samples, we demonstrate that the prevalence and abundance of LAB species in stool samples is generally low and linked to age, lifestyle, and geography, with *Streptococcus thermophilus* and *Lactococcus lactis* being most prevalent. Moreover, we identify genome-based differences between food and gut microbes by considering 666 metagenome-assembled genomes (MAGs) newly reconstructed from fermented food microbiomes along with 154,723 human MAGs and 193,078 reference genomes. Our large-scale genome-wide analysis demonstrates that closely related LAB strains occur in both food and gut environments and provides unprecedented evidence that fermented foods can be indeed regarded as a possible source of LAB for the gut microbiome.

[1] Department of Agricultural Sciences, University of Naples Federico II, Portici, Italy. [2] Task Force on Microbiome Studies, University of Naples Federico II, Naples, Italy. [3] CIBIO Department, University of Trento, Trento, Italy. [4] Teagasc Food Research Centre, Moorepark, Fermoy, Ireland. [5] APC Microbiome Ireland, Cork, Ireland. ✉email: danilo.ercolini@unina.it

For several decades, lactic acid bacteria (LAB) have been among the most extensively studied microorganisms. LAB have a fundamental role in different biological processes and ecosystems, especially with respect to fermented foods. The microbiology of fermentations has been extensively studied for over a century and the ability to transform raw materials into edible products with defined characteristics dates back to thousands of years as a strategy of food preservation[1,2]. Industrial fermentations are based on selected cultures that are used as starters or adjuncts to guarantee specific metabolic activities along with quality, reproducibility, and safety. On the other hand, artisanal processes do not usually involve defined starter cultures and the LAB available in the raw materials, or sourced from a previous manufacturer, lead the fermentation. Food-associated LAB have been studied mainly from the perspective of their fermentation performances and phenotypic properties, and knowledge on such properties has recently increased, thanks to intense genome sequencing of LAB strains[3,4].

Apart from their contributions to food quality and safety, LAB have attracted considerable interest due to their potentialities to add functional properties to certain foods or as supplements. Functional foods are designed to deliver additional benefits over their basic nutritional values and contribute to human health[5]. In this regard, several LAB species and strains have been recognized as probiotics, i.e., "live microorganisms that confer a health benefit on the host when administered in adequate amounts[6]." Importantly, many LAB species also enjoy a generally recognized as safe status.

Despite the extensive literature focusing on characterizing LAB in food, it is still not fully understood how they interact with the human gut microbiome[5]. Ingested LAB need to first survive the physical and chemical barriers of the gut, before competing with hundreds of different species, and finally being able to exert their beneficial effects. Indeed, LAB are regarded as components of the transient gut microbial community, coming from the external environment and with food representing the main source, which interacts daily with the longer term members of the gut microbiome[7]. Despite this general view, it is still not known to what extent components of the food microbiome are actively transferred to become part of the gut microbiome and what role they play in this complex environment. Depending on the specific food, technology of production, and fermentation process, fermented foods can harbour several LAB species and strains, and are natural sources of live microorganisms that are consumed daily across all human populations, and that can potentially interact with the gut microbiome. Despite this, the degree to which LAB species and strains not explicitly regarded as probiotics can be transferred to the gut has been largely under-explored. In addition, no studies have been conducted to assess the distribution of LAB in the global population, a gap that may

be bridged by taking advantage of the growing availability of high-throughput sequencing data.

In this study, we perform a large-scale genome-wide analysis of publicly available and newly sequenced food and human metagenomes to investigate the prevalence and diversity of LAB species with a view to identifying links between gut and food microbiomes. We find that LAB species occur with variable prevalence and generally low abundance in the human gut. Such prevalence is affected by age and lifestyle. LAB species identified in food only partially match those in the gut. Comparative genomics suggest an overall food origin for the gut strains.

## Results

**Large-scale meta-analysis on food and human microbiomes.** We performed a large-scale meta-analysis on microbiomes from food sources and human body sites to investigate the prevalence and diversity of LAB species in the human microbiome and their overlap with species and strains found in food. To achieve this goal, we considered 303 food metagenomes (152 publicly available and 151 obtained in this study) (11 datasets; Table 1 and Supplementary Data 1) that we curated in this study, which corresponded to different types of fermented foods and beverages[8–14]. In addition, we considered 9445 human metagenomes from 47 public datasets spanning multiple body sites (84% from the gut), age categories, countries, and lifestyles, which we retrieved from recent meta-analyses[15,16].

**Variable prevalence of LAB in the human gut.** We considered reference-based taxonomic profiles[17] of all 9445 human metagenomes[15,16] (see "Methods") and focused specifically on LAB species in this study (Supplementary Data 2). We detected 152 species belonging to the Lactobacillales order occurring in at least one of the metagenomes with a relative abundance >0.01%. Among them, we identified 70 species belonging to the LAB group and restricted the following analysis to the 30 of them having a prevalence >0.1% in the human gut (see "Methods"). These represented mainly species (spanning *Lactobacillus*, *Lactococcus*, *Leuconostoc*, *Streptococcus*, and *Weissella* genera) of potential food origin, including bacteria occurring in probiotic supplements, in addition to typically non-food origin species such as *Lactobacillus mucosae*, *Lactobacillus ruminis*, and *Lactobacillus salivarius* (Fig. 1). The two most prevalent species in the gut were *Streptococcus thermophilus* (prevalence 31.2%, i.e., present at >0.01% relative abundance in 31.2% of the gut metagenomes) and *Lactococcus lactis* (16.3%), both commonly found in dairy products (Fig. 1, Supplementary Fig. 1, and Supplementary Data 3). Multiple *Lactobacillus* species of predominantly food origin were detected at lower prevalence (3–5%) and comprised *Lactobacillus casei/paracasei*, *Lactobacillus delbrueckii*, *Lactobacillus fermentum*,

**Table 1 Summary of the analysed food metagenomic datasets.**

| Study | Type of food | # Samples | Accession number | Reference |
|---|---|---|---|---|
| BertuzziAS_2018 | Surface ripened cheese | 42 | PRJEB15423 | 8 |
| Escobar-ZepedaA_2016 | Mexican ripened cheese | 1 | PRJNA286900 | 9 |
| LeechJ_2019 | Fermented food | 58 | PRJEB35321 | This study |
| MacoriG_2019 | Cheese | 77 | PRJEB32768 | This study |
| MilaniC_2019 | Parmesan cheese | 2 | PRJNA482503 | 10 |
| PasolliE_2019 | Yoghurt and dietary supplement | 16 | PRJNA603575 | This study |
| PfeferT_2018 | Cheese | 36 | PRJNA430402 | – |
| QuigleyL_2016 | Continental type cheese | 10 | PRJEB6952 | 11 |
| WalshAM_2016 | Milk kefir | 18 | PRJEB15432 | 12 |
| WalshAM_2017 | Nunu | 20 | PRJEB20873 | 13 |
| WolfeBE_2014 | Smear ripened cheese | 23 | mgp3362 | 14 |

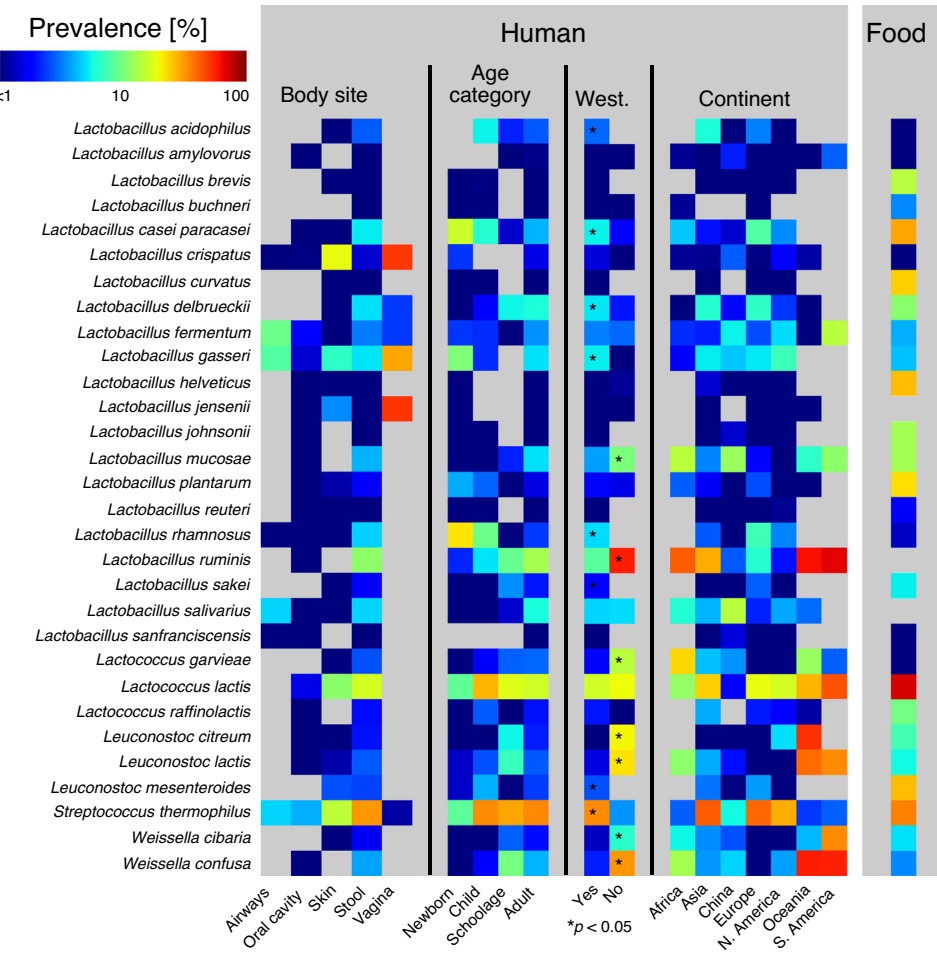

**Fig. 1 Average prevalence of LAB species from human and food microbiomes.** We report the 30 LAB species having a prevalence >0.1% in the human gut. Values are obtained from 9445 publicly available human metagenomes and stratified by multiple host conditions (i.e., body site, age category, westernized lifestyle, and continent). Age category, westernized lifestyle, and continent statistics refer to stool samples only. Food results are obtained from 303 food metagenomes. Numbers and p-values (Fisher's test, false discovery rate correction) in Supplementary Figs. 1–4 and Supplementary Data 4. Relative abundances in Supplementary Data 2 and 3.

and *Lactobacillus rhamnosus*). Non-food origin bacteria were also identified at remarkable levels such as *Lb. ruminis* (11.0%), *Lb. salivarius* (4.7%), and *Lb. mucosae* (4.0%). Although prevalence was variable, average relative abundance (computed on positive samples only) of single species was generally rather low (<2%), including the case of the two most prevalent species *S. thermophilus* (0.6%) and *Lc. lactis* (0.4%). Exceptions (rel. ab. >2%) were verified for *Lactobacillus amylovorus*, *Lactobacillus brevis*, and *Lactobacillus buchneri*, which however rarely occurred (prev. <1%).

Strong age-related patterns were verified for some of the species prevalent in gut samples ($N = 7907$) (Fig. 1, Supplementary Fig. 2, and Supplementary Data 4). *S. thermophilus* increased in prevalence from newborns (8.4%) to adults (33.7%, $p < 1e - 40$), with comparable average abundance. This may reflect the increase in consumption of yoghurts and other dairy products that can be sources of *S. thermophilus*[18]. A similar pattern was observed for *Lb. delbrueckii* ($p < 1e - 10$) and the non-food origin species *Lb. mucosae* ($p < 1e - 10$), *Lb. ruminis* ($p < 1e - 20$), and *Lb. salivarius* ($p < 1e - 10$), which suggests their gut colonization later in age. Also, *Lc. lactis* had higher prevalence in adults (15.8%) than newborns (8.6%, $p < 1e - 6$), with its detection in only one infant cohort originating from Estonia, Finland, and Russia[19]. Other lactobacilli were more prevalent and abundant in newborns such as *Lb. casei/paracasei* ($p < 1e - 20$ with respect to

adults), *Lactobacillus gasseri* ($p < 1e - 7$), *Lactobacillus plantarum* ($p < 1e - 4$), and *Lb. rhamnosus* ($p < 1e - 70$). These species have also been detected in human breast milk[20], suggesting their possible transmission from mother to infant through breastfeeding, as previously reported for *Lb. plantarum*[21]. Notably, these species were not found to be vertically transmitted from other mother's body sites[22].

Overall, we found that LAB are a subdominant component of the gut microbiome, although several species exhibited non-negligible contributions. More specifically, we identified 21 LAB occurring with prevalence >1% and 18 with relative abundance >0.5% when detected in the gut. It is reasonable to hypothesize that those species may be short- or long-term colonizers of the human microbiome.

**Occurrence and abundance of LAB is linked to lifestyle.** We then stratified the gut metagenomes in terms of host lifestyles (Fig. 1, Supplementary Fig. 3, and Supplementary Data 4), which revealed variations in prevalence and abundance between westernized and non-westernized populations for multiple species. Higher prevalence in westernized populations was observed for six lactobacilli, mostly of food origin, such as *Lactobacillus acidophilus* ($p < 1e - 6$), *Lb. casei/paracasei* ($p < 1e - 4$), *Lb. delbrueckii* ($p < 0.01$), *Lb. gasseri* ($p < 1e - 6$), *Lb. rhamnosus* ($p < 1e - 9$), and *Lactobacillus sakei* ($p < 1e - 3$). By contrast, *Lb.*

*mucosae* ($p < 1e - 8$) and *Lb. ruminis* ($p < 1e - 100$) that do not occur in food were more prevalent in the non-westernized cohorts. Despite different patterns in terms of prevalence, all lactobacilli were on average more abundant in the westernized populations. Among the other genera, *S. thermophilus* was highly prevalent in the westernized cohorts ($p < 1e - 50$). Higher prevalence in the non-westernized group was observed for *Lactococcus garvieae* ($p < 1 - e30$) in addition to multiple heterofermentative species such as *Leuconostoc citreum* ($p < 1e - 70$), *Leuconostoc lactis* ($p < 1e - 60$), *Weissella cibaria* ($p < 1e - 10$), and *Weissella confusa* ($p < 1e - 100$), which is consistent with their widespread prevalence in raw vegetables[23] that are likely consumed in such populations. In fact, non-western populations usually have hunter–gatherer diet and lifestyle, which is recognized to be characterized by high consumption of tubers, drupes, roots, and fruits[24,25]. Indeed, it was also reported that the!Kung and the Hadza, two non-Western African populations, still obtain 60–80% and 50–65% of their diet from plant foods, respectively[26].

We further grouped metagenomes by host country of origin (see "Methods") and identified more subtle geographical variations (Fig. 1 and Supplementary Fig. 4). Overall, food-associated lactobacilli were most prevalent and abundant in Europe, were less so in Asia and North America, and were almost absent in China (kept distinct from the other Asian countries due to its large sample size) and in the non-westernized populations. The higher prevalence in European cohorts was significant ($p < 0.05$) for *Lb. casei/paracasei* (8.0%), *Lb. delbrueckii* (6.6%, with a similar value in Asia), and *Lb. rhamnosus* (7.1%). Exceptions were *Lb. gasseri*, having comparable prevalence in continents including westernized cohorts, and *Lb. fermentum*, more prevalent in North America, South America, and China, with the latter observation being consistent with its widespread occurrence in Chinese fermented foods[27]. Non-food lactobacilli were not prevalent in Europe. *Lb. mucosae* exhibited high prevalence (>10%) in Africa, China, and South America, with comparable abundance across the globe. A similar trend was verified for *Lb. ruminis*, although with higher prevalence in non-westernized cohorts, whereas the presence of *Lb. salivarius* was distinctive for the Chinese population ($p < 0.01$). Among the other genera, *Lc. lactis* exhibited high prevalence across the entire globe (ranging from 11.5% in Africa to 44.4% in South America) with the sole exception of China (1.7%). *S. thermophilus* reached high prevalence in Asia (41.5%), Europe (39.6%), and North America (28.1%), but was much less prevalent in the Chinese (5.6%) and non-westernized (<3%) cohorts.

## LAB species from food only partially match those in the gut.
We established genome level links between the microorganisms populating the human microbiome and those found in food by integrating the genomes reconstructed from a set of 9445 human metagenomes with those from the set of 303 food metagenomes that we generated, collected, and curated in this work (Table 1 and Supplementary Data 1). More specifically, we considered 303 metagenomic samples spanning 11 datasets and coming from different types of cheese ($N = 191$), multiple fermented foods ($N = 58$), nunu ($N = 20$), milk kefir ($N = 18$), and yoghurt and dietary supplements ($N = 16$). We applied a validated[16,28] computational pipeline that combined single-metagenome assembly, contig binning, and genome quality control to reconstruct de novo metagenome-assembled genomes (MAGs) from the set of food metagenomes (see "Methods"). We generated a total of 666 food MAGs (completeness > 50% and contamination < 5%) of sufficient quality according to previous recommendations[29]. These MAGs from food were integrated with the set of 154,723 MAGs that we retrieved from the 9445 human metagenomes

using the same assembly-based pipeline[16] and with the set of 193,078 reference genomes (available in GenBank as of March 2019). This resulted in a total of 348,467 genomes that were clustered at 5% genetic distance based on whole-genome nucleotide similarity estimation and recapitulated in species-level genome bins (SGBs, i.e., clusters of genomes spanning 5% genetic diversity, see "Methods"). The 666 food MAGs were grouped into 171 SGBs (Supplementary Data 5 and 6), which we discuss below on the basis of their occurrence in food samples and human gut (Fig. 2a, b).

Most of the food MAGs (349, 52.4%) belonged to SGBs also found in the human gut, with 265 of them associated with twenty of the thirty LAB species discussed previously (Fig. 2a top panel and Supplementary Fig. 5). The species most reconstructed from food sources was *Lc. lactis* ($N = 90$ MAGs), with 86 MAGs extracted from cheese. Sixty MAGs were associated with *S. thermophilus*, the majority of them was reconstructed from cheese and yoghurt, and five additional genomes were extracted from different fermented foods such as wagashi, beetroot kvass, ryazhenka, ruž'a, and labne. A consistent number of MAGs was also retrieved from *Lactobacillus helveticus* (33 MAGs from cheese), *Lactobacillus curvatus* (14 MAGs from cheese and 1 from sauerkraut), *Lb. delbrueckii* (11 MAGs from cheese or yoghurt in addition to single genomes from dietary supplement and tofu), *Leuconostoc mesenteroides* (5 MAGs from nunu and single genomes from bread kvass, ginger beer, milk kefir, beetroot kvass, ruž'a, and cheese), and *Lb. casei/paracasei* (4 MAGs from cheese, 2 MAGs from dietary supplements, and 2 MAGs from water kefir). We also extracted four MAGs of *Lb. mucosae*, a typical non-food microorganism that is usually found in the intestine of pigs or other animals[30], and which we instead reconstructed from different fermented foods such as kimchi, kombucha vinegar, agousha, and sauerkraut.

We identified 17 additional non-LAB SGBs having MAGs from both food and human metagenomes, for a total of 84 food MAGs (12.6%; Fig. 2a bottom panel) and spanning three phyla (namely Actinobacteria, Firmicutes, and Proteobacteria). Some of these may be microbial contaminants in the food chain that can arise from different sources including animal, feed, and soil[31,32]. The SGB with the most MAGs ($N = 16$) was that containing *Streptococcus equinus* and *Streptococcus infantarius* genomes, two species usually found in the rumen[33] but occasional pathogens for humans[34], and which we found in African fermented foods[13].

The majority of the food SGBs (134 out of 171), accounting for 317 MAGs (47.6%), did not exhibit an overlap with human MAGs, likely representing species unable to reach the colon or characterized by low prevalence and abundance in the human gut (Fig. 2b). Among them, 71 SGBs (53.0%; comprising 225 MAGs) contained at least one reference genome (kSGBs; Fig. 2b left panel). The most prevalent food-specific species was *Brevibacterium linens* (24 MAGs), which was reconstructed from multiple cheese types (i.e., surface ripened[8], smear ripened[14], hard, and tomme). Food-specific SGBs also included *Staphylococcus saprophyticus* (13 MAGs), *Glutamicibacter arilaitensis* (12 MAGs), and 58 MAGs from 21 LAB species spanning 6 families, the most prevalent being *Lc. lactis* subsp. *cremoris*. This set of MAGs and reference genomes showed a >5% genetic distance from *Lc. lactis* subsp. *lactis* genomes[35], which we kept as a separate SGB (ID 7985) and found to be prevalent in both food and human metagenomes, in contrast to *Lc. lactis* subsp. *cremoris*, which was only detected in food metagenomes. Similarly, *Lactococcus raffinolactis* was divided into two SGBs, with human and food MAGs grouped in the SGBs 7989 and 7991, respectively.

Out of the 134 SGBs not overlapping with human MAGs, 63 SGBs (47%; comprising 92 MAGs) consisted of MAGs

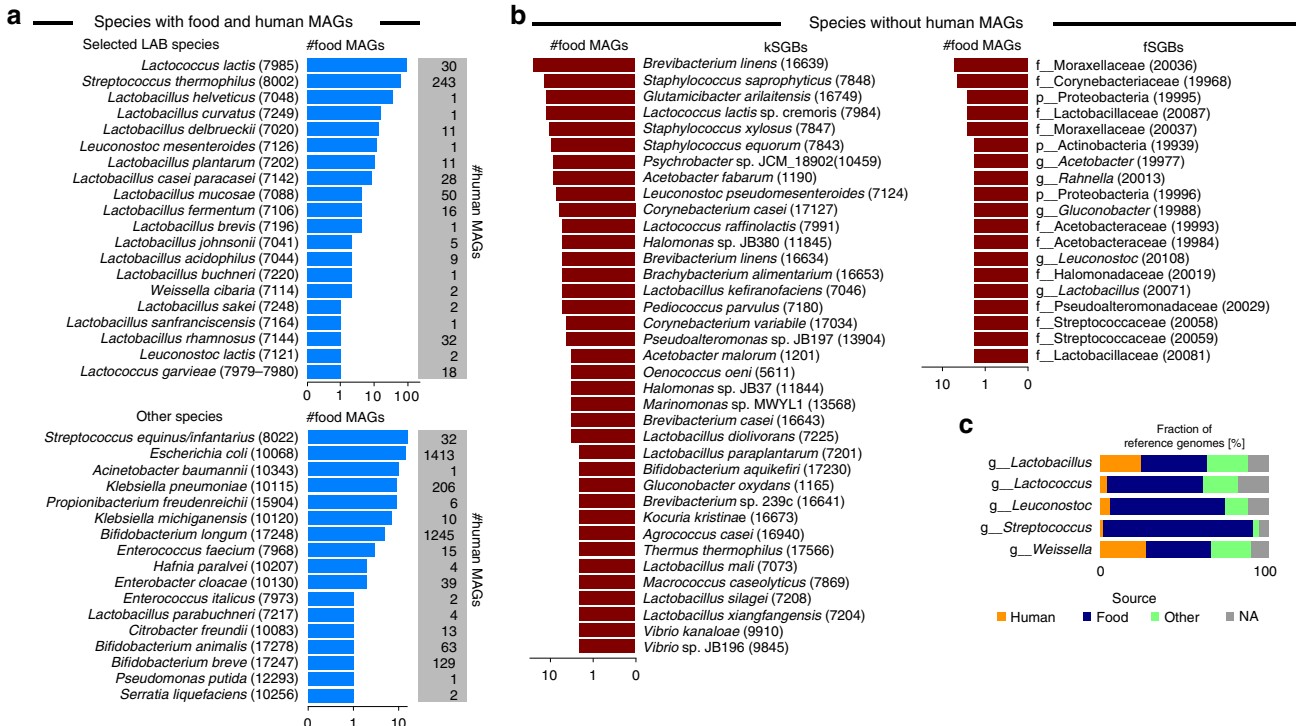

**Fig. 2 Microbial genomes reconstructed from food metagenomes. a** Most prevalent species-level genome bins (SGBs) in 666 MAGs reconstructed from 303 food metagenomes and overlapping with human MAGs (i.e., found in at least one of the 154,723 human MAGs). Numbers in parenthesis represent the SGB IDs. **b** Most prevalent food SGBs not overlapping with human MAGs. kSGBs denote SGBs with at least one reference microbial genome, whereas fSGBs identify newly assembled SGBs from food metagenomes only. X-axes for **a** and **b** are in logarithmic scale. **c** Fraction of reference genomes per source type for the 30 selected LAB species and grouped by genera (the same plot at species-level is reported in Supplementary Fig. 6). Raw data in Supplementary Data 6 and 7.

reconstructed in this study from food metagenomes without any reference genomes. These represented new species currently not represented in public repositories (Fig. 2b right panel), of which only 12 were assigned to known genera, and which should be targeted for cultivation-based analysis.

The set of genomes reconstructed and the SGBs identified in this study and that we made publicly available (see "Methods") facilitated a more in-depth comparative genomics analysis.

**Comparative genomics suggests a food origin for the gut strains.** Within the available set of MAGs and reference genomes, we performed strain-level comparative genomic analysis for the set of 348,467 genomes previously described and comprising 193,078 reference genomes, 154,723 human MAGs, and 666 food MAGs. The 2859 genomes (including 1042 MAGs) associated with the thirty LAB species of interest were kept for comparative genomics purposes. To inform the comparative analysis, we retrieved and manually curated the source types for all genomes (see "Methods") and grouped MAGs and reference genomes in three categories: human, food, and other. Genomes for which this information was missing were labelled as NA (7.9% of genomes; Fig. 2c, Supplementary Fig. 6, and Supplementary Data 7).

Overall, two-thirds of the reference genomes came from food (43.8%) and human sources (21.0%). The group of genomes from strains not isolated from foods or humans (22.8%) comprised 67 genomes from probiotics and dietary supplements in addition to 347 genomes mainly coming from animal sources. The proportions of species assigned to the different source types was quite variable across species, with a general under-representation of human genomes corresponding to LAB that were prevalent in non-westernized cohorts (Fig. 2c and Supplementary Fig. 6). This

reflected the overall scarce availability of genome from isolates for a substantial fraction of the non-pathogenic, commensal members of the human microbiome as recently highlighted[16,36,37]. Reference genomes from human samples were surprisingly almost absent in the case of prevalent species such as *Lc. lactis* (with only one reference genome from the vagina and one MAG from the gut) and *S. thermophilus* (with only one MAG from the gut). The absence of good reference genomes in public repositories prevented the comparison of food and human strains until now, which we aimed to overcome in the present study through an extensive comparative genomics analysis.

*S. thermophilus* was the species of LAB most frequently reconstructed from metagenomes (243 human and 60 food MAGs; Fig. 3a), an observation consistent with its high prevalence from mapping-based taxonomic profiling (Fig. 1). Comparative genomics, also including 44 reference genomes, did not highlight food-specific or gut-specific sub-clades, suggesting that food can be regarded as the main source of this species in the human microbiome. *S. thermophilus* also appeared to be a quite genetically diverse species both in food and human sources with MAGs reconstructed from Asian gut metagenomes enriched in a specific clade (Clade A, Fig. 3a, $p < 1e - 10$). *Lb. delbrueckii* was not prevalent in the gut, and the only two subspecies found in human samples were subsp. *lactis* and subsp. *bulgaricus* (Fig. 4a). Human MAGs of both subspecies clustered together with food MAGs and isolates, again indicating food as the most likely source of this species in the gut. On the other hand, subsp. *delbrueckii*, subsp. *sunkii*, and subsp. *jakobsenii* were found in food, but never reconstructed from the gut. Although *Lb. rhamnosus* was the LAB species for which the greatest number of genomes corresponding to human isolates ($N = 105$) was available, we collected only 32 human MAGs, which is in

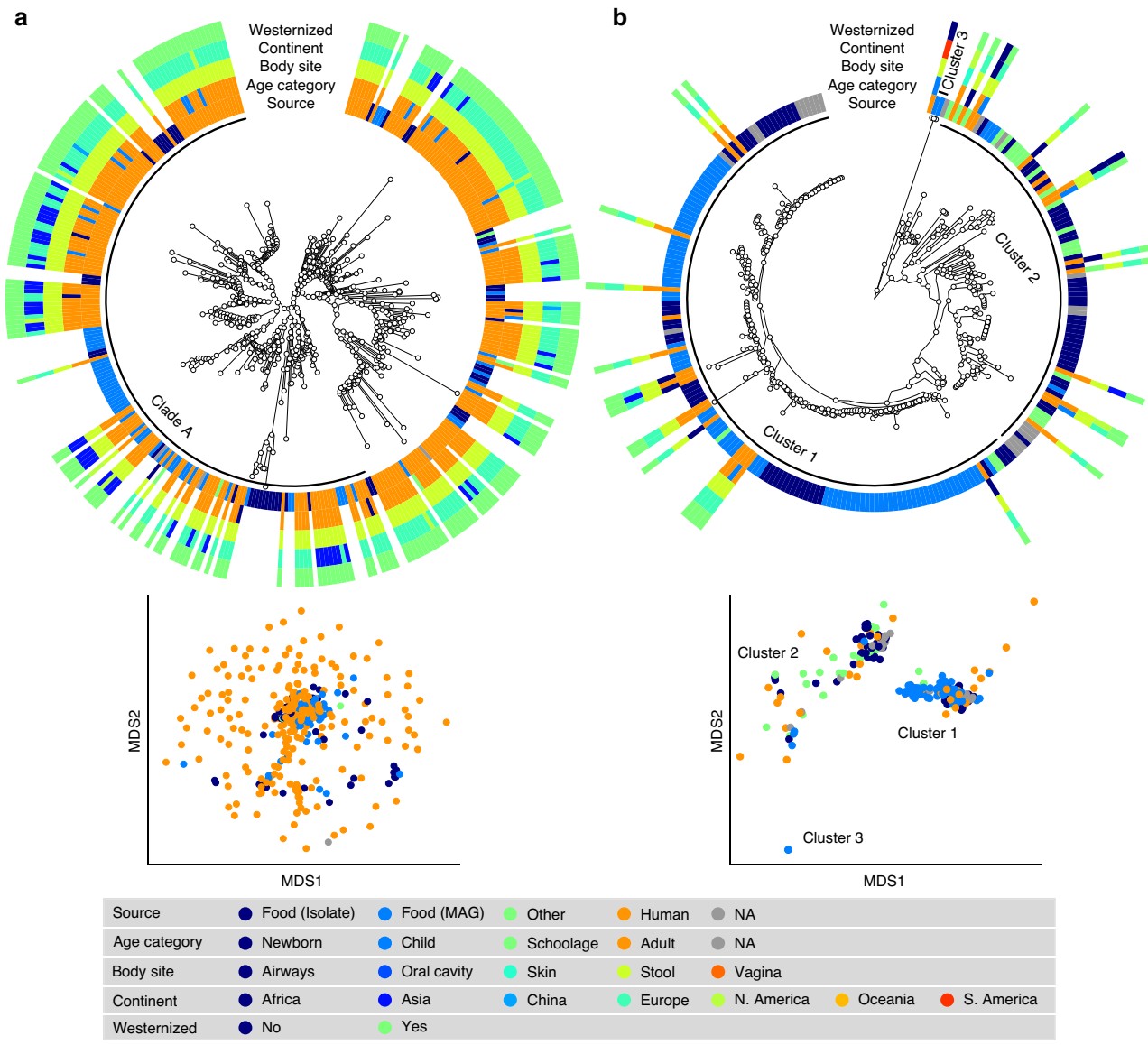

**Fig. 3 Comparative genomic analysis of the two most prevalent LAB identified in the human gut microbiome. a** *S. thermophilus* is a genetically diverse species both in food and human sources with MAGs reconstructed from Asian gut metagenomes enriched in Clade A ($p < 1e - 10$). **b** *Lc. lactis subsp. lactis* is formed by three main clusters: Cluster 1 exhibits an overall low diversity and includes mostly food genomes related to cheese and dairy fermentation; Cluster 2 is dominated by environmental and raw vegetable products and more diverse human MAGs; Cluster 3 includes only two MAGs from nunu. Phylogenetic trees were built on species-specific marker genes and report five different metadata. Multidimensional scaling (MDS) on average nucleotide identity (ANI) distance is coloured with source information.

agreement with its low prevalence and abundance in the gut (Fig. 4b). We identified a specific cluster including 17% of the *Lb. rhamnosus* human genomes that included the reference genome associated with the *Lb. rhamnosus* strain GG (LGG), which may be due to recent consumption of commercial products due to its wide use in probiotic supplements[38].

The highest number of food MAGs was obtained for *Lc. lactis* ($N = 90$, Fig. 3b). We refer here to subsp. *lactis*, whereas subsp. *cremoris* was associated with 12 food MAGs but never reconstructed from human metagenomes. *Lc. lactis* subsp. *lactis* formed two distinct clusters including both food and human genomes. The first cluster included 63% of the genomes, exhibited an overall low diversity (<0.8% genetic distance between closest genome pairs), and included all the food genomes related to cheese and dairy fermentation. The second cluster was more diverse, dominated by environmental and raw vegetable products, and included the only MAG from human skin and the three gut

MAGs from non-westernized cohorts. An additional cluster containing two genomes from nunu[13] was never found in humans and exhibited a >3% genetic diversity from all other genomes. Such results highlighted the overall importance of conducting strain-level analysis on the food-gut axis, depicted here by the identification of two main clusters in the human gut associated with different food sources (i.e., one from cheese and dairy fermentation, and the other one from environmental and raw vegetables products). Strains of these clusters are likely characterized by differences in functional traits and potential interaction with the host that deserve to be investigated in future studies.

The SGB 7142 ($N = 216$, Fig. 4c), labelled *Lb. casei/paracasei*, included reference genomes identified as both *Lb. casei* and *Lb. paracasei*, which, as recently highlighted, can be used interchangeably[39]. Within the combined species, we detected two main clusters, both of which occurred in food and human

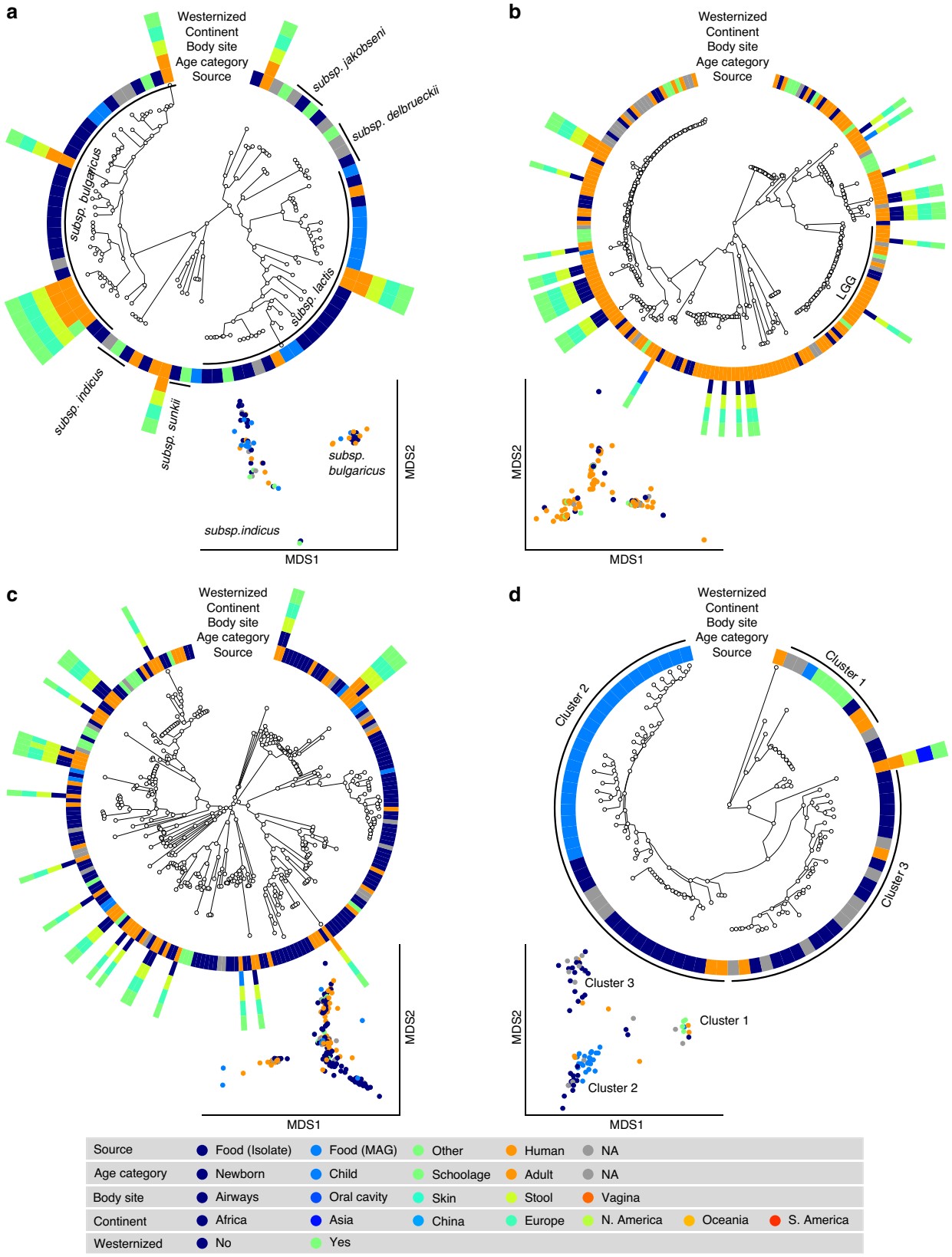

| Source | ● Food (Isolate) | ● Food (MAG) | ● Other | ● Human | ● NA |
|---|---|---|---|---|---|
| Age category | ● Newborn | ● Child | ● Schoolage | ● Adult | ● NA |
| Body site | ● Airways | ● Oral cavity | ● Skin | ● Stool | ● Vagina |
| Continent | ● Africa | ● Asia | ● China | ● Europe | ● N. America ● Oceania ● S. America |
| Westernized | ● No | ● Yes | | | |

samples. The major cluster contained 86% of the available genomes, including all the dietary supplement strains and the majority (86%) of the human MAGs. Consistent with its low abundance (Fig. 1), only seven reference genomes and a single MAG were reconstructed from human samples for *Lb. helveticus* (Fig. 4d). We identified three main subspecies, all occurring in

both food and human sources. One cluster included all the dietary supplement strains, whereas genomes coming from food were predominantly spread across the other two groups.

Despite the high number of collected genomes ($N = 369$), *Lb. plantarum* was scarcely prevalent (1.8%) and abundant (av. 1.2%) in the gut (Fig. 1), which was reflected by only 11 MAGs being

**Fig. 4 Comparative genomic analysis of relevant lactobacilli found in both food and human microbiomes. a** *Lb. delbrueckii* is not prevalent in the gut, and the only two subspecies found in both food and human samples are subsps. *lactis* and. *bulgaricus*. Subsps. *delbrueckii, sunkii,* and *jakobsenii* are found in food, but never reconstructed from the gut. **b** *Lb. rhamnosus* exhibits the greatest number of genomes from human isolates but is scarcely reconstructed from metagenomes. A specific cluster identifies the LGG strain. **c** *Lb. casei/paracasei* includes reference genomes identified as both *Lb. casei* and *Lb. paracasei*. We detect two main clusters both occurring in food and human samples. **d** *Lb. helveticus* exhibits three main clusters, with Cluster 1 including all the dietary supplement strains (source in green), while food genomes are predominantly spread across the other two groups. Phylogenetic trees were built on species-specific marker genes and report five different metadata. Multidimensional scaling (MDS) on average nucleotide identity (ANI) distance is coloured with source information.

reconstructed from human microbiomes (Supplementary Fig. 7). All of these belonged to the main cluster (96% of the total genomes) associated with subsp. *plantarum*. A separate cluster was identified as subsp. *argentoratensis*, which was found in both food and human isolates but never reconstructed from metagenomes. The occurrence of multiple subspecies within the same SGB was also observed for eight additional LAB, i.e., *Lb. brevis, Lb. fermentum, Lactobacillus johnsonii, Lactobacillus reuteri, Lb. sakei, L. lactis, L. mesenteroides,* and *W. cibaria,* (Supplementary Fig. 7). On the other hand, *Lc. garvieae* was spread into two different SGBs, with one comprising human MAGs from both westernized and non-westernized populations and the other only from non-westernized cohorts (Supplementary Fig. 7). No genomes from food samples were collected at all for *Lactobacillus crispatus, Lb. gasseri, Lactobacillus jensenii, Lb. ruminis,* and *Lb. salivarius* (excluding a single isolate from ground beef). The non-food species *Lb. ruminis* and *Lb. salivarius* were quite prevalent in the gut with 145 and 42 MAGs reconstructed from human metagenomes, respectively (Supplementary Fig. 7). For both species, isolate and MAGs extracted from the gut were distinct from genomes isolated from other animal microbiomes, which suggested long-term adaptation of these species to the human gut. We also identified a specific *Lb. salivarius* cluster associated with dietary supplement strains, which was found in a couple of saliva samples but never in the human gut.

**LAB occurrence in non-human primates is affected by captivity.** We finally considered the set of 203 publicly available gut metagenomes from non-human primates (NHPs) that was recently retrieved, curated, and processed with the same pipeline employed in this study[28]. It comprised 22 host species from 14 different countries in five continents. Among the 2985 reconstructed MAGs, we found that only 46 of them (1.6%) were assigned to the Lactobacillales order (Supplementary Data 8), which suggested an overall low prevalence and abundance of LAB in the NHP gut microbiome. We found strong differences between MAGs retrieved from wild NHPs and those extracted from NHPs living in captivity. Wild NHPs generated 29 MAGs of LAB, with 66% of them associated with new species not available in public repositories and never found in human metagenomes, therefore likely representing bacteria peculiar to the NHP gut microbiomes. Ten MAGs were instead associated with kSGBs, with only five of them belonging to LAB species found also in human gut metagenomes such as *Lc. garvieae* (N = 3), *Lc. lactis,* and *W. cibaria.* Comparative genomics analysis highlighted that the strains harboured in NHPs were quite different from those reconstructed from human microbiomes (Supplementary Fig. 8). Interestingly, the three MAGs of *Lc. garvieae* resembled more the strains found in non-westernized human populations in terms of nucleotide identity. No MAGs from lactobacilli were extracted at all from wild NHPs. A very different situation was observed in captive NHPs (Supplementary Fig. 8), in which the 17 MAGs were exclusively reconstructed from kSGBs associated with multiple *Lactobacillus* species, i.e., *Lb. acidophilus, Lactobacillus animalis* (N = 2), *Lb. johnsonii* (N = 4), *Lb. mucosae* (N = 2), *Lb.*

*reuteri* (N = 5), and *Lb. salivarius* (N = 3). Strains of *Lb. reuteri* and *Lb. salivarius* found in NHPs were distinct from those extracted from human and food sources, which suggested possible host adaptation mechanisms. A stronger overlap among NHPs, human, and food MAGs was instead observed for the other species and likely linked to the sharing of strains due to the exposition of NHPs living in captivity to human-like environments and diets[40].

## Discussion

In this study, we showed that food is likely the major source of LAB in the human gut microbiome. This was accomplished by conducting a large-scale meta-analysis that integrated taxonomic profiling and comparative genomics from almost ten thousand metagenomes from human and food sources in addition to reference genomes from public repositories. We focused the analysis on the thirty LAB that exhibited a prevalence >0.1% in the human gut, which resulted mainly in species of potential food origin, including LAB occurring in probiotic supplements, in addition to non-food origin species such as *Lb. mucosae, Lb. ruminis,* and *Lb. salivarius.* The comparative genomics suggested that closely related strains are present in both food and gut microbiome. Although such evidence does not exclude the possibility of other potential sources of LAB strains in the gut, we believe that being fermented foods the principal ecological niche for LAB in nature, our results support the hypothesis that food is the major source of LAB for the gut microbiome. While we considered the currently available taxonomic nomenclature, a substantial reclassification of the genus *Lactobacillus* into 25 novel genera enclosing the current *Lactobacillus* species was recently proposed[41]. The new *Lactobacillus* genus incorporates only the species included in the *Lb. delbrueckii* group.

We found an overall limited amount of LAB in the gut in terms of prevalence and relative abundance; however, several species exhibited non-negligible contributions that deserve attention for potential probiotic potentials. There was no evident correlation between prevalence and relative abundance of the different LAB species in the human samples. The most prevalent LAB species was *S. thermophilus.* Its role as a gut microbiome member is questioned. However, the mechanisms and metabolic features that lead to it being regarded as a candidate probiotic species have been studied and debated, especially in terms of resistance to gastrointestinal barriers and potential positive health effects[42]. Beyond being one of the two LAB widely employed for yoghurt making, *S. thermophilus* is also employed as starter cultures for many cheeses characterized by a thermophilic fermentation. Continuous exposure to *S. thermophilus* through cheese and yoghurt consumption can be a likely explanation of its prevalence in human gut samples as resulted in this study.

We detected a remarkable prevalence in the gut also for *Lc. lactis,* which is widespread in cheeses produced by mesophilic fermentation. Albeit recognized as a transient member of the gut community, higher levels of this species were found in buttermilk consumers[43]. In addition, strains of *L. lactis* have been shown to survive the gastrointestinal stress and this species can be

considered to potentially convey health benefits by antimicrobial activity through bacteriocin production against clostridia, to boost the immune system, and to be potentially used as a vehicle of interesting beneficial properties such as antimicrobial activity[44,45].

The prevalence of LAB in the human gut was strongly affected by lifestyle[46], intended here as possible consumption of fermented foods that are characteristics of specific geographical regions. Unfortunately, direct associations of genomic data with dietary patterns could not be achieved as dietary records documenting systematic food consumption in the human public cohorts considered were not available. Minor associations between gut microbiota and consumption of plant fermented foods were very recently found within the American gut cohort. A few LAB species were linked to fermented plant food consumers and included *Lb. acidophilus*, *Lb. brevis*, *Lactobacillus kefiranofaciens*, *Lactobacillus parabuchneri*, *Lb. helveticus*, and *Lb. sakei*. Interestingly, the authors highlighted that the stool detection of LAB may be a useful tool to verify the reliability of self-reported dietary information on fermented foods consumption[47].

In our study, LAB species widely occurring in dairy products and yoghurt, such as *S. thermophilus* and lactobacilli, were more prevalent in westernized populations, whereas the heterofermentative *Leuconostoc* and *Weissella*, likely carried as part of the epiphytic microbiota of raw vegetables[23], fermented vegetables[48], and cereal-based fermented foods[49] were more common in the non-westernized cohorts. We could speculate that this pattern was linked to the habitual consumption of foods and diets that were characteristics of the specific geographical areas. For example, non-westernized populations that have a higher consumption of raw plants and plant based fermented foods were enriched in heterofermentative cocci LAB, whereas the very low prevalence of *Lc. lactis* and *S. thermophilus* in multiple Chinese cohorts reflects the low consumption of dairy products by the Chinese population[50].

We conducted an extensive comparative genomic analysis by integrating reference genomes and MAGs from human, food, and environmental sources. This opportunity was previously prevented even for prevalent species such as *S. thermophilus* and *Lc. lactis* due to the lack of reference genomes acquired from human sources in public repositories. We identified a general overlap among genomes from food and gut sources, which suggested again food as the main source of LAB in the human gut. To this end, we conducted a preliminary analysis devoted to evaluate potential differences in functions of strains between food and gut sources, that we limited to *Lc. lactis* and *S. thermophilus* due to their large number of MAGs reconstructed in this study (see "Methods"). We found 266 (247 in food) and 323 (275 in food) differently prevalent genes ($p < 0.05$) for *Lc. lactis* and *S. thermophilus*, respectively, after removing genes encoding for unidentified functions or occurring redundantly in both food and gut groups (differently prevalent sugar metabolism genes are listed in Supplementary Data 9). However, such differences did not suggest remarkable potential functional differences between food and gut genomes, which was consistent with the comparative genomics and phylogenetic results shown in Fig. 3. At the same time, we identified an increase of unannotated genes in the gut genomes for both species, which agreed with the scarcity of reference genomes from human sources in public repositories. This may reflect further differences of strains found in the human gut that are currently unexplored due to the incompleteness of available functional databases[51]. Functional differences may suggest a possible adaptation of the food LAB to the gut environment. However, such mechanisms of adaptations cannot occur in strains that are part of a transient microbiome and would only take place for those LAB that more stably colonize the gut

environment. This opens the need to conduct new analyses focused on the isolation of these microorganisms from the gut and their more in-depth functional characterization, also based on phenotypic traits. Different patterns were observed for typical non-food origin species such as *Lb. ruminis* and *Lb. salivarius*. By comparing human genomes with those found in other environments including animal microbiomes, we identified a strong adaptation of these species to the human gut, which suggested that these species are more specific and persistent for the human host (Supplementary Fig. 7).

Some of the analysed LAB exhibited distinct groups with human and food genomes clustering together, which indicated the presence in the gut of different strains potentially coming from different food sources. For example, the genomes of *Lb. delbrueckii* reconstructed from the gut appeared to cluster in two main groups associated with subsp. *bulgaricus* and subsp. *lactis*, which were representative of LAB in yoghurt and cheese, respectively. Multiple subclusters were identified also in *Lb. rhamnosus*, with only 17% of the reconstructed human MAGs corresponding to the strain GG largely used in probiotic supplements. These species, along with others such as *Lb. casei*, *Lb. plantarum*, and *Lb. reuteri* have been largely explored due to their probiotic potential. However, their general low prevalence and abundance in the human gut suggested that they are unlikely to be long-term residents of the gut microbiota. However, we used only fecal samples as representative of the gut microbiome, while such species maybe more tightly adhered to the gut epithelium and therefore less detectable in stool specimen[52].

Finally, we highlight the importance of considering computational approaches such as those exploited in this paper. Strain-level genome comparison is fundamental to track the resilience and persistence of probiotic LAB in the human gut and can be a useful approach to be adopted in clinical trials aimed at evaluating the efficacy of microbial strains for gut health. In addition, the same methodologies can be considered to evaluate the prevalence and resilience of non-food microorganisms that are currently studied as candidate for next generation probiotics. Such knowledge and approaches can be useful for an informed design of functional foods, conveying health benefits upon daily consumption beyond their nutritional value. Several functional foods are enriched with probiotic microbial strains and their fate in pre-clinical and clinical trials can be efficiently and reliably monitored by culture-independent genome reconstruction and comparison to help assessing both their efficacy as probiotics and the quality of the functional food.

The interest in LAB will keep the scientific community active in studies of their genomics and evolution. Some of the LAB species occurring in the gut can surely arise from the consumption of fermented foods or probiotic preparations. However, efforts in research and isolation of LAB from human specimen would be desirable in the future in order to have further evidence on their specific genomic features that may better reflect adaptation to the complex gut ecosystem.

## Methods

**Publicly available and newly acquired food metagenomes**. We considered and curated public datasets from fermented food metagenomes in addition to food metagenomes newly sequenced in this study. In total we put together 303 samples spanning 11 datasets and coming from different types of cheese ($N = 191$), fermented foods ($N = 58$), nunu ($N = 20$), milk kefir ($N = 18$), and yoghurt and dietary supplements ($N = 16$)[8–14]. More information is detailed in Table 1. Additional information on the collected food metagenomes is available in Supplementary Data 1.

**Publicly available human metagenomes**. In addition, we considered publicly available metagenomic datasets corresponding to the human microbiome. More specifically, we included 47 human microbiome datasets totalling 9,445

metagenomes and 4.2e11 Illumina reads as done in ref. [16] (17 metagenomes that were left out due to technical issues in ref. [16] were included here by marginally expanding the original set of 9428 metagenomes). Overall, the samples were acquired from six major body sites: the gut by stool sampling ($N = 7907$), oral cavity ($N = 785$), skin ($N = 508$, including from the anterior nares), airways ($N = 151$), vagina ($N = 86$), and breast milk ($N = 8$, data not included in figures). These samples covered 31 countries that were grouped by continent as follows: Africa (MDG: Madagascar, TZA: Tanzania), Asia (BGD: Bangladesh, BRN: Brunei, IDN: Indonesia, ISR: Israel, KAZ: Kazakhstan, MNG: Mongolia, MYS: Malaysia, SGP: Singapore), China (CHN, which we kept separated from the other Asian countries due to its large sample size), Europe (AUT: Austria, DEU: Germany, DNK: Denmark, ESP: Spain, EST: Estonia, FIN: Finland, FRA: France, GBR: Great Britain, HUN: Hungary, ISL: Iceland, ITA: Italy, NLD: The Netherlands, NOR: Norway, RUS: Russia, SVK: Slovakia, SWE: Sweden), North America (CAN: Canada, USA: United States), Oceania (FJI: Fiji), and South America (PER: Peru). The samples were also categorized as corresponding to westernized ($N = 8850$) and non-westernized ($N = 595$) lifestyles[16]. More specifically, westernization is a complex process that occurred during the last few centuries and that involved lifestyle changes compared with populations prior to the modern era. Such changes include increased hygiene and sanitized environments, introduction of antibiotics and other drugs, increased high-calorie high-fat dietary regimes, enhanced exposure to pollutants, and reduced contact with wildlife and domesticated animals. We adopt westernized and non-westernized as umbrella terms to depict populations that differ by the majority of the aforementioned factors even though this definition comprises heterogeneous populations. Finally, these metagenomes spanned multiple age categories: newborns ($N = 711$, <1 year of age), children ($N = 802$, age ≥ 1 and <12 years), school age individuals ($N = 215$, age ≥ 12 and <19 years), and adults ($N = 7669$, age ≥ 19 years). Despite curation efforts, age category metadata corresponding to 48 samples could not be sourced. These manually curated metadata are available in the Supplementary Data 2 and in the *curatedMetagenomicData* package[15].

**Taxonomic profiling of food and human metagenomes.** Quantitative taxonomic profiling was applied on the 9,445 human metagenomes and the 303 food metagenomes by applying MetaPhlAn2[17] with default parameters. MetaPhlAn2 estimates relative abundances of microbial species using the pre-generated ~1 M unique clade-specific marker genes identified from ~17,000 reference genomes (~13,500 bacterial and archaeal, ~3500 viral, and ~110 eukaryotic). Taxonomic profiles along with associated metadata information are available in Supplementary Data 2. We detected 152 species belonging to the Lactobacillales order occurring in at least one of the metagenomes with a relative abundance >0.01%. Among them, we identified 70 species belonging to the LAB group (i.e., species belonging to *Lactobacillus*, *Lactococcus*, *Leuconostoc*, and *Weissella* genera in addition to *S. thermophilus*), and restricted the rest of the analysis to the 30 of them having a prevalence >0.1% in the human gut. Taxonomic profiles of these 30 species are reported in Fig. 1 and Supplementary Figs. 1–4. Prevalence was computed by thresholding relative abundance at 0.01%. Average relative abundance was computed on positive samples only.

**Metagenome-assembled genomes reconstruction.** Taxonomic profiling was coupled with the reconstruction of microbial genomes directly from metagenomes. The approach that we validated in[16] was applied here to reconstruct MAGs MAGs from food metagenomes. More specifically, single-sample metagenomics assemblies were generated with metaSPAdes[53] (version 3.10.1; default parameters) or IDBA-UD[54] (version 1.1.3; default parameters). Contigs longer than 1,000 nt were binned with MetaBAT2[55] (version 2.12.1; option "-m 1500"). Quality control with CheckM (v. 1.0.7)[56] yielded 666 medium-quality food MAGs (completeness > 50% and contamination <5%). These newly reconstructed MAGs were then considered within the human MAG catalogue totalling 154,723 MAGs reconstructed from the 47 human datasets considered in this study[16].

**Clustering of genomes into species-level genome bins.** The 155,389 MAGs described in the previous section were integrated with the set of 193,078 reference genomes available in GenBank as of March 2019. This resulted in a total of 348,467 genomes that were clustered into SGBs following the procedure proposed in[16]. Genomes were clustered with average linkage at 5% genetic distance based on whole-genome nucleotide similarity estimation using Mash (v. 2.0; option "-s 10000" for sketching)[57]. The 666 food MAGs were grouped by this procedure into 171 SGBs: 108 SGBs (comprising 574 MAGs) contained at least one reference genome or human MAG (kSGBs), while a further 63 SGBs (comprising 92 MAGs) consisted only of genomes reconstructed in this study from food metagenomes (fSGBs). Summaries of the newly generated MAGs and SGBs are available in Fig. 2a, b and Supplementary Data 5 and 6.

**Metadata curation for selected LAB species.** We considered the 30 selected LAB species shown in Fig. 1 for comparative genomics purposes. Among the 348,467 genomes described in the previous section, 2859 genomes (comprising 1042 MAGs) were included in SGBs containing at least one reference genome assigned to these 30 species and were kept for further analyses. We retrieved and manually curated the source type in all cases. For reference genomes, the source of isolation was extracted from the NCBI portal or from related publications. Genomes were

grouped in three categories based on the source type: "human," "food," and "other." Genomes for which this information was missing were labelled as "NA" ($N = 226$, 7.9% of the cases). More information relating to these 2859 genomes is available in Supplementary Data 7.

**Reconstruction of phylogenetic structure.** Phylogenies were built using the newly developed PhyloPhlAn 3.0 package that extends the original PhyloPhlAn2 version[58]. Each SGB-specific phylogeny (Fig. 3) was based on the set of species-specific marker genes that can be retrieved in PhyloPhlAn 3.0 with the command phylophlan2_setup_database.py. The number of marker genes for each SGB is summarized in Supplementary Data 10. This departs from the default option in using the 400 universal markers available in PhyloPhlAn 3.0 and guarantees a higher resolution of the built phylogenies. The parameters were set as follows "--diversity low --fast --min_num_marker 50", which indicated that genomes mapping less than 50 markers were discarded from the phylogeny. External tools embedded in PhyloPhlAn 3.0 were run with their specific options as follows:

- blastn (version 2.6.0 +;[59]) with parameters "-outfmt 6 -max_target_seqs 1000000"
- mafft (version 7.310;[60]) using the "L-INS-i" algorithm and with parameters "--anysymbol --auto"
- trimal (version 1.2rev59;[61]) with parameter "-gappyout"
- FastTree (version 2.1.9;[62]) with parameters "-mlacc 2 -slownni -spr 4 -fastest -mlnni 4 -no2nd -gtr -nt"
- RAxML (version 8.1.15;[63]) with parameters "-p 1989 -m GTRCAT -t <phylogenetic tree computed by FastTree >"

Phylogenetic trees (Figs. 3 and 4) were visualized with GraPhlAn[64]. In addition, multidimensional scaling plots (Figs. 3 and 4, and Supplementary Figs. 4 and 5) were built on the whole-genome Average Nucleotide Identity distances computed with FastANI[65].

**Functional analysis and statistical significance.** The set of genomes (MAGs and reference genomes) considered in this study was annotated with Prokka (v. 1.12;[66]) using default parameters. Proteins inferred by Prokka were then processed with Roary[67] (v. 3.11; option '-i 90') to generate the presence–absence binary matrix on the core and accessory genes. Gene enrichment within human and food genomes was determined by considering only MAGs and reference genomes having completeness >80% in order to avoid possible biases coming from highly incomplete genomes and by taking into account genes present in at least 5% and <95% of the genomes. Statistical significance was tested through Fisher's test with false discovery rate correction for multiple hypothesis testing.

**Reporting summary.** Further information on research design is available in the Nature Research Reporting Summary linked to this article.

## Data availability

The raw data for the food metagenomes are available in NCBI-SRA under the BioProjects PRJEB6952 [https://www.ncbi.nlm.nih.gov/bioproject/PRJEB6952], PRJEB15423 [https://www.ncbi.nlm.nih.gov/bioproject/PRJEB15423], PRJEB15432 [https://www.ncbi.nlm.nih.gov/bioproject/PRJEB15432], PRJEB20873 [https://www.ncbi.nlm.nih.gov/bioproject/PRJEB20873], PRJEB32768 [https://www.ncbi.nlm.nih.gov/bioproject/PRJEB32768], PRJEB35321 [https://www.ncbi.nlm.nih.gov/bioproject/PRJEB35321], PRJNA286900 [https://www.ncbi.nlm.nih.gov/bioproject/PRJNA286900], PRJNA430402 [https://www.ncbi.nlm.nih.gov/bioproject/PRJNA430402], PRJNA482503 [https://www.ncbi.nlm.nih.gov/bioproject/PRJNA482503], PRJNA603575 [https://www.ncbi.nlm.nih.gov/bioproject/PRJNA603575], and in MG-RAST under the Project mgp3362 [https://www.mg-rast.org/linkin.cgi?project=mgp3362].

The taxonomic profiles with associated metadata from the human metagenomes are available in the *curatedMetagenomicData* package[15]. The MAGs from human metagenomes are available at http://segatalab.cibio.unitn.it/data/Pasolli_et_al.html. The newly reconstructed MAGs from food metagenomes are available at http://www.tfm.unina.it/DATA001-2020-Pasolli.

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

## Acknowledgements

This study was developed within the project MASTER (Microbiome Applications for Sustainable food systems through Technologies and Enterprise). The project has received funding from the European Union's Horizon 2020 research and innovation programme under grant agreement No 818368. This manuscript reflects only the authors' views and the European Commission is not responsible for any use that may be made of the information it contains. The study has been also supported by the JPI HDHL-INTIMIC - Knowledge Platform of Food, Diet, Intestinal Microbiomics and Human Health, granted by the Italian Ministry of Agricultural, Food and Forestry and Tourism Policies with ID 790.

## Author contributions

Conception: D.E. and E.P. Formal analysis: E.P., F.D.F., I.E.M., A.W., J.L., and F.C. Resources and funding acquisition: D.E., N.S., and P.C. Writing–original draft, E.P. and D.E. Writing–review and editing, E.P., D.E., N.S., and P.C. All authors read and approved the final manuscript.

## Competing interests

The authors declare no competing interests.
