## [Peer Review File · Nature Communications]

Reviewers' comments:

Reviewer #1 (Remarks to the Author):

Overall, the paper is coherently written and presents a good case of using MAGs to study microbial prevalence in food metagenome at the strain level, given the lack of good reference genomes from food and human isolates. The analysis is clear, and the relevant statistics are provided where appropriate. The additional dataset of 151 metagenomes from known food sources that contain LAB, such as fermented food, cheese and yogurt, is also a valuable resource for identifying novel strains of LAB in food.

However, some of the suggestions that LAB in the gut comes directly from food are tenuous because, as the authors correctly pointed out, "direct associations of genomic data with dietary patterns could not be achieved as dietary records documenting systematic food consumption in the human public cohorts considered were not available." While comparative analysis found clusters that contain both human genome and food MAG + isolates, the clusters suggest at best that closely related strains are present in both food and gut. The analysis does not provide a causal relationship between the two nor eliminate other potential sources of the strains. Further studies that include dietary patterns of humans from different age groups and geography are necessary to establish the causal relationship between the gut and food.

Below are some specific feedbacks:

- Add a column to Fig 1 showing the abundance of different species for different food metagenome (grouped by types of food).
 - o It'll help make the case of why taxonomic classification alone is insufficient for associating LAB in food and the gut.
- It can be useful to add a column of p-values on whether there is significant difference of in prevalence between sub-groups to identify LAB species of interest.
- Are there any trends between prevalence and relative abundance?
- Abundance was not addressed significantly in the paper, unlike prevalence. Perhaps the subfigure showing abundances could be moved to supplementary materials instead.
- Add a column to Fig.2A showing the #human MAGs that correspond to each LAB species.

- In Fig 3A's ordinate plot, there is a large overlap between *S. thermophilus* genomes from human and food at the center of the plot. While the authors claim that genomes from human sources are diverse (which the plot reflects), the figure suggests that there could be at least one cluster. On the right half of the dendrogram, there seem to be more genomes from Asian human sources compared to the left half.
- When the text/caption discusses clusters in the comparative analysis, the corresponding subfigures in Fig 3 and 4 should highlight the clusters visually by highlighting the leaves in a separate color, or by drawing arcs like in Fig. 4A.
- In the discussion section, the authors mention that they did a preliminary analysis to evaluate the differences of functions of strains between food and gut sources. Perhaps they could add the data as supplementary information.

Line 144 – widespread 'presence'

Line 186 - (75.9%) is ambiguous, since it is 75.9% of 349, not 666. Omit it.

Line 187 – 'mostly' to 'most'

Line 190 – rephrase 'apart their widespread'

Line 360 – 'which is' widespread

Line 512 -- 2,859 genomes (comprising 1,042 MAGs) 'were' included 'in' SGBs containing at least one reference genome assigned to these 30 species and were kept for further analyses.

Line 518 – Do you mean 226 cases of NA, which is 12.4%?

Reviewer #2 (Remarks to the Author):

Pasoli et al. characterize in this study the prevalence and abundance of food-derived and autochthonous gut lactic acid bacteria (LAB). The last authors are well-experienced microbiome scientists with a very good track record on genome-based analyses using metagenome-assembled genomes (MAGs). In this study, the authors reconstructed 303 MAGs from fermented food microbiomes and integrated those with almost 350,000 human MAGs and reference genomes. Looking at more than 9000 human samples of different origin and context, including body sites, age, and country, the authors conducted the genome-wide analysis on the prevalence, abundance and diversity of 30 LAB species. In my view, this is a technically outstanding piece of work addressing a question which has been discussed over the past 15 years. The authors confirmed at the molecular level that food-derived LAB exist in the human gut ecosystem with mostly low abundance. The authors mentioned that they focus mostly on lactobacilli, lactococci, *Leuconostoc* ssp., *Str.*

thermophilus and Weisella with a prevalence >0.1%. In Fig. 2B, the authors show that most of the food metagenomes do not overlap with the human system. Here I wondered why bifidobacteria (also LAB) are not seen in the human gut? To my knowledge bifidobacteria are more prevalent and usually also more abundant compared to lactobacilli. Do the assembled metagenomes cover bifidobacterial species diversity? At one point the authors discuss the fact that *Str. thermophilus* is the most abundant LAB in the human ecosystem and that others may not survive the gastric transit. This is in my view a poor argument since *Str. thermophilus* is a very acid sensitive species. One rationale might be that exposure to *Str. thermophilus* is likely very high because yoghurt consumption provides continuous input. It would be an interesting experiment to expose individual very low for *Str. thermophilus* to yoghurt and to measure kinetics of appearance and potentially disappearance. In summary, I have little criticism apart from the few questions and I think this study will provide a reference for the food and probiotic research.

Reviewer #3 (Remarks to the Author):

Pasolli et al. present a well-written report on lactic acid bacterial genomes from the human microbiome and fermented foods. The authors describe which lactic acid bacteria tend to be found in either sample type and present some groups of lactic acid bacteria that can be present in food as well as the human gut microbiome. They subject the genomes of these bacteria to comparative genomics analyses. Finally, they also analyse the prevalence and relative abundance of these genomes in human samples, in attempt to relate these measures to host factors. While most of the data has been taken from an earlier metaanalysis by the same authors and the study is descriptive in nature, the results are novel especially in the size of the data base and the depth of presentation. Due to potential of lactic acid bacteria as probiotic agents, the author's aim to better understand the seeding of the human microbiome from different food sources is timely and interesting for human microbiome and nutrition specialists alike. However, the study would gain considerably from presenting also functional genetic information. I enjoyed reading the clear data presentation, also the methodology is laudably described.

As you will see from my comments below, I have few concerns about the study itself. I have two comments on the scope of the study, however:

- 1) My main problem with the "genome-wide analyses" is the total absence of any functional data. It would be immensely interesting to know more about the metabolic potential of the described bacteria. Marker genes for the fermentation of specific substrates have been described in LABs for more than a decade. I find the study incomplete without a description of at least the different sugars that could be fermented by the different strains and how this relates to their distribution and also phylogeny.

2) With the focus of the title and arguably the most interesting angle for the analysis being on the relationship of food and gut LAB, I would ask whether passages on the skin/vaginal microbiomes are necessary for the story or more of a distraction? Similarly the section on non-human primate microbiomes does not add much in this sense (although I found it very interesting that wild primates have no LABs at all). I would leave it up to authors to decide whether to tighten the scope in this regard.

Data presentation/statistics:

- In the paragraph starting in line 85, it is not clear to me what the relative abundance data are (e.g. <2% in l. 101): are these cohort average? averages from samples with the species being present? medians? maximal values?

- same question for the paragraph starting in l. 105 (if you want to keep it).

- as it plays a prominent role in some of the conclusions, it would be good to define westernization for the scope of this paper, so readers don't have to refer to citation 16. I did, and I am confused as to the different numbers of samples in the earlier paper and the present paper - surely you have an explanation for this?

- on a related note, what is the evidence for a higher consumption of raw vegetables in the non-westernized cohorts? furthermore, are you certain that there was a high consumption of kimchi (a Korean dish) in these cohorts, as claimed in l. 144?

- l. 212: what does "most prevalent" refer to, i.e. where/in which SGB/foods was *B. linens* most prevalent? please clarify this sentence

- l. 332: what measure is the resemblance based on? nucleotide identity?

Figures / legends:

- figure 1: is the scale percentage?

- figure 2: I would prefer the scale of the bars in A and B to not be linear; especially in the bottom part of A, as well as B, it's not obvious from the axis label that it is - also it's kind of wrong to have the bars extend below 1. Potentially, the whole figure might be more appropriately presented as a table.

- figure 3 and 4: please indicate in the figure legends what the cladograms are based on. If I understood it correctly, the trees are built based on marker gene distances and not whole-genome ANIs? this would be very important to mention.

Language and structure:

- abstract: I feel the abstract could do with more conclusion and less methodological data. I found it difficult to discern the main message other than "We analysed a lot of data" (which you did and that's great).

- l. 137: replace "resulted" with "were"

- l. 144 and l. 190: insert "occurrence" or "presence" behind "widespread"

- l. 172: remove superfluous "a"

- l. 185: insert "food" before "MAGs" for clarity

- l. 187: replace "mostly reconstructed" with "most often/commonly reconstructed"

- l. 190: replace "apart" with "despite"

- l. 205: replace "most reconstructed SGB" with "SGB with the most MAGs"

- l. 333: replace "scenario" with "situation"

- legend to figure 4: replace "included" in the last sentence with "including"

Reviewer #1

Overall, the paper is coherently written and presents a good case of using MAGs to study microbial prevalence in food metagenome at the strain level, given the lack of good reference genomes from food and human isolates. The analysis is clear, and the relevant statistics are provided where appropriate. The additional dataset of 151 metagenomes from known food sources that contain LAB, such as fermented food, cheese and yogurt, is also a valuable resource for identifying novel strains of LAB in food.

However, some of the suggestions that LAB in the gut comes directly from food are tenuous because, as the authors correctly pointed out, “direct associations of genomic data with dietary patterns could not be achieved as dietary records documenting systematic food consumption in the human public cohorts considered were not available.” While comparative analysis found clusters that contain both human genome and food MAG + isolates, the clusters suggest at best that closely related strains are present in both food and gut. The analysis does not provide a causal relationship between the two nor eliminate other potential sources of the strains. Further studies that include dietary patterns of humans from different age groups and geography are necessary to establish the causal relationship between the gut and food.

Response

First we would like to thank the Reviewer for appreciating our work and research idea. Indeed, the lack of dietary information represents a limit that prevents the possibility to establish a causal relationship between occurrence of LAB in food and gut. As the Reviewer highlights, the comparative genomics suggests that closely related strains are present in both food and gut. While it is technically true that such evidence does not exclude the possibility of other potential sources of LAB strains, the Reviewer will concur that this is unlikely because fermented foods are the principal ecological niche for LAB in nature. Since we overall agree with the Reviewer’s concern, we pointed out such limitations more clearly in the discussion of our results. First, we clearly explained the basis of our conclusion from the beginning of the discussion section:

(line 350) “The comparative genomics suggested that closely related strains are present in both food and gut microbiome. While such evidence does not exclude the possibility of other potential sources of LAB strains in the gut, we believe that being fermented foods the principal ecological niche for LAB in nature, our results support the hypothesis that food is the major source of LAB for the gut microbiome.”.

In addition, we acknowledged the lack of dietary information in our cohorts as a limit of the study and also discussed evidence on the link between the consumption of (only plant) fermented foods and microbiota signatures that was recently published:

(line 376) “Unfortunately, direct associations of genomic data with dietary patterns could not be achieved as dietary records documenting systematic food consumption in the human public cohorts considered were not available. Minor associations between gut microbiota and consumption of plant fermented foods were very recently found within

the American gut cohort. A few LAB species were linked to fermented plant food consumers and included *Lb. acidophilus*, *Lb. brevis*, *Lb. kefirifaciens*, *Lb. parabuchneri*, *Lb. helveticus* and *Lb. sakei*. Interestingly, the authors highlighted that the stool detection of LAB may be a useful tool to verify the reliability of self-reported dietary information on fermented foods consumption ⁴⁵.

Reference 45: “Taylor, Bryn C., Franck Lejzerowicz, Marion Poirer, Justin P. Shaffer, Lingjing Jiang, Alexander Aksenov, Nicole Litwin, et al. 2020. “Consumption of Fermented Foods Is Associated with Systematic Differences in the Gut Microbiome and Metabolome.” *mSystems* 5 (2). <https://doi.org/10.1128/mSystems.00901-19>.”

Below are some specific feedbacks:

- **Add a column to Fig 1 showing the abundance of different species for different food metagenome (grouped by types of food). It'll help make the case of why taxonomic classification alone is insufficient for associating LAB in food and the gut.**

Response

We thank the Reviewer for the suggestion, we think that this is a valuable addition to the results. We added the prevalence of the LAB species in the food metagenomes in the last column of **Figure 1**:

Figure 1. Average prevalence of LAB species from human and food microbiomes. We report the 30 LAB species having a prevalence greater than 0.1% in the human gut. Values are obtained from 9,445 publicly available human metagenomes and stratified by multiple host conditions (i.e., body site, age category, westernized lifestyle, and continent). Age category, westernized lifestyle, and continent statistics refer to stool samples only. Food results are obtained from 303 food metagenomes. Numbers and p-values in **Figure S1** and **Table S4**. Relative abundances in **Table S2** and **Table S3**.

The LAB prevalence was only reported in aggregated form and not split by food category as the available metagenomes for food products were not evenly distributed in the different food categories.

We removed the relative abundance information from **Figure 1** according to another comment by Reviewer 1 and reported later in this rebuttal letter. Relative abundances from human metagenomes are still summarized in **Figure S1**, while the same information from food metagenomes is reported in the newly added **Table S3**:

“Table S3. Average prevalence and relative abundance of the 30 LAB species across the 303 food metagenomes estimated with MetaPhlAn2.”

- **It can be useful to add a column of p-values on whether there is significant difference of in prevalence between sub-groups to identify LAB species of interest.**

Response

We added a new table in the supplementary material (**Table S4**) that reports the p-values among the different comparisons conducted on the MetaPhlAn2 taxonomic profiles for all the four categories (i.e., age category, body site, continent, and westernized lifestyle). This is the caption:

Table S4. P-values associated with the Fisher’s test (after FDR correction) applied on the presence/absence of the 30 LAB species determined by MetaPhlAn2 spanning age, body site, continent, and westernized lifestyle categories.

We added this information also in **Figure 1** for the westernized lifestyle comparison (please see the **Figure 1** reported in the previous reply).

- **Are there any trends between prevalence and relative abundance?**

Response

Overall, there was no actual correlation between prevalence and abundance. We have some parts in the text discussing this aspect. Some prevalent species occurred at low abundance levels (such as *Lc. lactis* or *S. thermophilus*): (line 100) “*While prevalence was variable, average relative abundance (computed on positive samples only) of single species was generally rather low (<2%), including the case of the two most prevalent species S. thermophilus (0.6%) and Lc. lactis (0.4%).*” and (line 106) “*S. thermophilus increased in prevalence from newborns (8.4%) to adults (33.7%, $p < 1e-40$), with comparable average abundance*”. Some other species were much less prevalent but occurred at higher abundance values: (line 103) “*Exceptions (rel. ab. >2%) were verified for Lb. amylovorus, Lb. brevis, and Lb. buchneri, which however rarely occurred (prev. < 1%).*”

We also decided to clarify this in the discussion section:

(line 357) “**There was no evident correlation between prevalence and relative abundance of the different LAB species in the human samples.**”

- **Abundance was not addressed significantly in the paper, unlike prevalence. Perhaps the subfigure showing abundances could be moved to supplementary materials instead.**

Response

In agreement with the suggestion, we removed the abundance information from **Figure 1** and kept this information in the supplementary material (more specifically from human metagenomes in **Figure S1** and from food metagenomes in **Table S3**).

- **Add a column to Fig.2A showing the #human MAGs that correspond to each LAB species.**

Response

We modified **Figure 2A** by adding the number of human MAGs for each LAB species. This is the updated **Figure 2** that takes into account also some additional suggestions made by Reviewer 3:

Figure 2. Microbial genomes reconstructed from food metagenomes. A) Most prevalent species-level genome bins (SGBs) in 666 MAGs reconstructed from 303 food metagenomes and overlapping with human MAGs (i.e., found in at least one of the 154,723 human MAGs). **B)** Most prevalent food SGBs not overlapping with human MAGs. kSGBs denote SGBs with at least one reference microbial genome, whereas fSGBs identify newly assembled SGBs from food metagenomes only. X-axes for panels **A)** and **B)** are in logarithmic scale. **C)** Fraction of reference genomes per source type for the 30 selected LAB species and grouped by genera (the same plot at species-level is reported in **Figure S3**). Raw data in **Table S6** and **Table S7**.

• In Fig 3A's ordinate plot, there is a large overlap between *S. thermophilus* genomes from human and food at the center of the plot. While the authors claim that genomes from human sources are diverse (which the plot reflects), the figure suggests that there could be at least one cluster. On the right half of the dendrogram, there seem to be more genomes from Asian human sources compared to the left half.

Response

We thank the Reviewer for the nice suggestion. Indeed MAGs reconstructed from Asian gut metagenomes are enriched in a specific clade. We discuss this in the text as follows:

(line 257) "*S. thermophilus* also appeared to be a quite genetically diverse species both in food and human sources with MAGs reconstructed from Asian gut metagenomes enriched in a specific clade (Clade A, **Figure 3A**, $p < 1e-10$)."

We modified consequently **Figure 3A**, which is reported in the next reply.

- When the text/caption discusses clusters in the comparative analysis, the corresponding subfigures in Fig 3 and 4 should highlight the clusters visually by highlighting the leaves in a separate color, or by drawing arcs like in Fig. 4A.

Response

Thanks for the suggestion, we modified **Figure 3** and **Figure 4** by highlighting the clusters discussed in the text. These are the two updated figures:

Figure 3. Comparative genomic analysis of the two most prevalent LAB identified in the human gut microbiome. A) *S. thermophilus* is a genetically diverse species both in food and human sources with MAGs reconstructed from Asian gut metagenomes enriched in Clade A ($p < 1e-10$). B) *Lc. lactis* subsp. *lactis* is formed by three main clusters: Cluster 1 exhibits an overall low diversity and includes mostly food genomes related to cheese and dairy fermentation; Cluster 2 is dominated by environmental and raw vegetable products and more diverse human MAGs; Cluster 3 includes only two MAGs from nunu. Phylogenetic trees were built on species-specific marker genes and report five different metadata. Multidimensional scaling (MDS) on average nucleotide identity (ANI) distance is coloured with source information.

Figure 4. Comparative genomic analysis of relevant lactobacilli found in both food and human microbiomes. A) *Lb. delbrueckii* is not prevalent in the gut, and the only two subspecies found in both food and human samples are subsps. *lactis* and *bulgaricus*. Subsp. *delbrueckii*, *sunkii*, and *jakobsenii* are found in food, but never reconstructed from the gut. **B)** *Lb. rhamnosus* exhibits the greatest number of genomes from human isolates but is scarcely reconstructed from metagenomes. A specific cluster identifies the LGG strain. **C)** *Lb. casei/paracasei* includes reference genomes identified as both *Lb. casei* and *Lb. paracasei*.

We detect two main clusters both occurring in food and human samples. **D)** *Lb. helveticus* exhibits three main clusters, with Cluster 1 including all the dietary supplement strains (source in green), while food genomes are predominantly spread across the other two groups. Phylogenetic trees were built on species-specific marker genes and report five different metadata. Multidimensional scaling (MDS) on average nucleotide identity (ANI) distance is coloured with source information.

• **In the discussion section, the authors mention that they did a preliminary analysis to evaluate the differences of functions of strains between food and gut sources. Perhaps they could add the data as supplementary information.**

Response

In agreement with the Reviewer's suggestion an additional supplementary table (**Table S9**) was added reporting genes belonging to sugar metabolism pathways (as suggested by the Reviewer 3) that were found to be significantly different between food and gut genomes of *Lc. lactis* and *S. thermophilus*. We focused on these 2 latter species as they were the ones with the largest number of MAGs reconstructed from gut samples in this study. This is the caption:

Table S9. List of sugar metabolism genes found to be differently prevalent ($p < 0.05$) between food and human gut genomes in *S. thermophilus* and *Lc. lactis*.

This new material is introduced in the main text as follows:

(line 403) "We found 266 (247 in food) and 323 (275 in food) differently prevalent genes ($p < 0.05$) for *Lc. lactis* and *S. thermophilus*, respectively, after removing genes encoding for unidentified functions or occurring redundantly in both food and gut groups (differently prevalent sugar metabolism genes are listed in **Table S9**)."

Line 144 – widespread ‘presence’

Response

Thank you, this was modified in "widespread prevalence" (see line 139).

Line 186 - (75.9%) is ambiguous, since it is 75.9% of 349, not 666. Omit it.

Response

We removed "(75.9%)" from the text (see line 184).

Line 187 – ‘mostly’ to ‘most’

Response

Thank you, we fixed this typo (see line 185).

Line 190 – rephrase ‘apart their widespread’

Response

We rephrased the sentence as follows:

(line 187) “Sixty MAGs were associated with *S. thermophilus* (N = 60); the majority of them was reconstructed from cheese and yogurt, and five additional genomes were extracted from different fermented foods such as wagashi, beetroot kvass, ryazhenka, ruž’a, and labne.”

Line 360 – ‘which is’ widespread

Response

Thank you, we fixed this typo (see line 367).

Line 512 -- 2,859 genomes (comprising 1,042 MAGs) ‘were’ included ‘in’ SGBs containing at least one reference genome assigned to these 30 species and were kept for further analyses.

Response

Thank you, we fixed this (see line 536).

Line 518 – Do you mean 226 cases of NA, which is 12.4%?

Response

Yes, we mean 226 cases among 2,859 genomes. We clarified this in the text as follows:

(line 541) “Genomes for which this information was missing were labelled as “NA” (N = 226, 7.9% of the cases). More information relating to these 2,859 genomes is available in **Table S7**.”

Reviewer #2

Pasoli et al. characterize in this study the prevalence and abundance of food-derived and autochthonous gut lactic acid bacteria (LAB). The last authors are well-experienced microbiome scientists with a very good track record on genome-based analyses using metagenome-assembled genomes (MAGs). In this study, the authors reconstructed 303 MAGs from fermented food microbiomes and integrated those with almost 350,000 human MAGs and reference genomes. Looking at more than 9000 human samples of different origin and context, including body sites, age, and country, the authors conducted the genome-wide analysis on the prevalence, abundance and diversity of 30 LAB species. In my view, this is a technically outstanding piece of work addressing a question which has been discussed over the past 15 years. The authors confirmed at the molecular level that food-derived LAB exist in the human gut ecosystem with mostly low abundance.

Response

We would like to thank the Reviewer for appreciating our work and research effort.

The authors mentioned that they focus mostly on lactobacilli, lactococci, *Leuconostoc* ssp., *Str. thermophilus* and *Weisella* with a prevalence >0.1%. In Fig. 2B, the authors show that most of the food metagenomes do not overlap with the human system. Here I wondered why bifidobacteria (also LAB) are not seen in the human gut? To my knowledge bifidobacteria are more prevalent and usually also more abundant compared to lactobacilli. Do the assembled metagenomes cover bifidobacterial species diversity?

Response

Although they are often part of probiotic preparations, bifidobacteria are not part of the LAB group and are genetically distant from LAB. Indeed, they belong to the phylum Actinobacteria. We deliberately decided not to consider this group in our study because we wanted to specifically look at the link between fermented foods and gut, and bifidobacteria are hardly present in fermented foods. A similar analysis on bifidobacteria would be surely of interest for the scientific community, and yes the Reviewer is right, the genus *Bifidobacterium* occurs with an average prevalence of 87.0% in the stool human metagenomes considered in this study (i.e. occurred in 6,876 out of 7,907 gut metagenomes). However, as above said, this was outside the scope of this specific investigation.

At one point the authors discuss the fact that *Str. thermophilus* is the most abundant LAB in the human ecosystem and that others may not survive the gastric transit. This is in my view a poor argument since *Str. thermophilus* is a very acid sensitive species. One rationale might be that exposure to *Str. thermophilus* is likely very high because yoghurt consumption provides continuous input. It would be an interesting experiment to expose individual very low for *Str. thermophilus* to yoghurt and to measure kinetics of appearance and potentially disappearance.

Response

We thank the Reviewer for highlighting this. We pointed out that (see line 359) “the role of *S. thermophilus* as a gut microbiome member is questioned. However, the mechanisms and metabolic features that lead to it being regarded as a candidate probiotic species have been studied and debated, especially in terms of resistance to gastrointestinal barriers and potential positive health effects”. We agree with the reviewer’s idea and explanation, which we partly already had in the results section commenting the increased prevalence of *S. thermophilus* from newborn to adults (see line 107): “This may reflect the increase in consumption of yogurts and other dairy products that can be sources of *S. thermophilus*”. Moreover, in order to further discuss our results in light of the reviewer’s advice, we decided to add the suggested interpretation of the prevalence of *S. thermophilus* in the discussion section

(line 362) “Beyond being one of the two LAB widely employed for yogurt making, *S. thermophilus* is also employed as starter cultures for many cheeses characterized by a thermophilic fermentation. Continuous exposure to *S. thermophilus* through cheese and yogurt consumption can be a likely explanation of its prevalence in human gut samples as resulted in this study.”

In summary, I have little criticism apart from the few questions and I think this study will provide a reference for the food and probiotic research.

Response

We would like to thank the Reviewer very much again for appreciating our work and its potential impact in the field.

Reviewer #3

Pasolli et al. present a well-written report on lactic acid bacterial genomes from the human microbiome and fermented foods. The authors describe which lactic acid bacteria tend to be found in either sample type and present some groups of lactic acid bacteria that can be present in food as well as the human gut microbiome. They subject the genomes of these bacteria to comparative genomics analyses. Finally, they also analyse the prevalence and relative abundance of these genomes in human samples, in attempt to relate these measures to host factors. While most of the data has been taken from an earlier metaanalysis by the same authors and the study is descriptive in nature, the results are novel especially in the size of the data base and the depth of presentation. Due to potential of lactic acid bacteria as probiotic agents, the author's aim to better understand the seeding of the human microbiome from different food sources is timely and interesting for human microbiome and nutrition specialists alike. However, the study would gain considerably from presenting also functional genetic information. I enjoyed reading the clear data presentation, also the methodology is laudably described.

Response

We would like to thank the Reviewer for appreciating our work and research effort.

As you will see from my comments below, I have few concerns about the study itself. I have two comments on the scope of the study, however:

1) My main problem with the "genome-wide analyses" is the total absence of any functional data. It would be immensely interesting to know more about the metabolic potential of the described bacteria. Marker genes for the fermentation of specific substrates have been described in LABs for more than a decade. I find the study incomplete without a description of at least the different sugars that could be fermented by the different strains and how this relates to their distribution and also phylogeny.

Response

We agree with the Reviewer that data on metabolic potential can be a valuable addition to comparative genomics. We did not emphasize much this aspect as our preliminary analyses showed no interesting differences at functional levels between food and gut LAB genomes. However, a remarkable part of the discussion section was dedicated to this aspect, which can be summarized as follows (please see lines 400-416). *We performed a functional analysis for *Lc. lactis* and *S. thermophilus*, which were the 2 LAB species with the larger number of MAGs reconstructed from gut in this study. We found 266 (247 in food) and 323 (275 in food) differently prevalent genes ($p < 0.05$) for *Lc. lactis* and *S. thermophilus*, respectively, after removing genes encoding for unidentified functions or occurring redundantly in both food and gut groups. The differences did not suggest potential functional differences between food and gut genomes, which was consistent with the close similarity found by comparative genomics (Figure 3). Functional differences may be useful to understand possible adaptation mechanisms of the food LAB to the gut environment. However, such mechanisms would hardly occur in strains that are part of a transient microbiome and would only take place for those LAB that more stably colonize the gut.*

This opens the need to conduct new analyses focused on the isolation of LAB from the gut and their more in-depth functional characterization. In fact, we believe that most informative functional characterization of LAB strains would be phenotypic, based on biochemical assays. Once enough gut isolates will be available, a more focused study of their functional traits may shed light on the possible actual differences between food and gut strains (if any) and on their different potential activities in the two different ecological niches.

Finally, we followed the relevant Reviewer's suggestion and included a new supplementary table (**Table S9**) containing a list of sugar metabolism genes that were found as differently prevalent ($p < 0.05$) between food and human gut genomes in *S. thermophilus* and *Lc. lactis*. This is the caption:

Table S9. List of sugar metabolism genes found to be differently prevalent ($p < 0.05$) between food and human gut genomes in *S. thermophilus* and *Lc. lactis*.

This new material is introduced in the main text as follows:

(line 403) "We found 266 (247 in food) and 323 (275 in food) differently prevalent genes ($p < 0.05$) for *Lc. lactis* and *S. thermophilus*, respectively, after removing genes encoding for unidentified functions or occurring redundantly in both food and gut groups (differently prevalent sugar metabolism genes are listed in **Table S9**)."

2) With the focus of the title and arguably the most interesting angle for the analysis being on the relationship of food and gut LAB, I would ask whether passages on the skin/vaginal microbiomes are necessary for the story or more of a distraction? Similarly the section on non-human primate microbiomes does not add much in this sense (although I found it very interesting that wild primates have no LABs at all). I would leave it up to authors to decide whether to tighten the scope in this regard.

Response

We thank the Reviewer for this suggestion. Accordingly, we decided to remove comments on the prevalence and relative abundances of LAB in non-gut samples from the text. However, we decided to keep the prevalence data in **Figure 1** as we believe this does not take too much space and could be of interest for more researchers studying the whole human microbiome. Similarly, although we believe that the Reviewer is right about the lower impact of the prevalence of LAB in non-human primate compared to the rest of the results, we preferred to keep it as a part of our study. Indeed, we find it interesting to show the LAB occurrence in human and wild primates in comparison to those held in captivity, as they hold interesting cues on the effects of westernization on human microbiome composition.

Data presentation/statistics:

- In the paragraph starting in line 85, it is not clear to me what the relative abundance data are (e.g. <2% in l. 101): are these cohort average? averages from samples with the species being present? medians? maximal values?

Response

We modified the sentence as follows:

(line 100): “While prevalence was variable, average relative abundance (computed on positive samples only) of single species was generally rather low (<2%), including the case of the two most prevalent species *S. thermophilus* (0.6%) and *Lc. lactis* (0.4%).”

- same question for the paragraph starting in l. 105 (if you want to keep it).

Response

The paragraph was removed in this revised version, according to the reviewer’s suggestion.

- as it plays a prominent role in some of the conclusions, it would be good to define westernization for the scope of this paper, so readers don’t have to refer to citation 16.

Response

We added this information (from the citation 16) in the “Materials and Methods” section:

(line 480): “More specifically, westernization is a complex process that occurred during the last few centuries and that involved lifestyle changes compared to populations prior to the modern era. Such changes include increased hygiene and sanitized environments, introduction of antibiotics and other drugs, increased high-calorie high-fat dietary regimes, enhanced exposure to pollutants, and reduced contact with wildlife and domesticated animals. We adopt westernized and non-westernized as umbrella terms to depict populations that differ by the majority of the aforementioned factors even though this definition comprises heterogeneous populations.”

I did, and I am confused as to the different numbers of samples in the earlier paper and the present paper - surely you have an explanation for this?

Response

In the earlier paper (Pasolli et al, *Cell*, 2019), we considered 9,428 human metagenomes. There were 17 metagenomes that were left out due to technical issues and that were instead included in this new analysis. The change in terms of numbers is very marginal (0.18%), however we agree that this was not clear and therefore it was clarified in the “Materials and Methods” section:

(line 465): “More specifically, we included 47 human microbiome datasets totalling 9,445 metagenomes and 4.2e11 Illumina reads as previously described ¹⁶ (seventeen metagenomes that were left out due to technical issues in ¹⁶ were included here by marginally expanding the original set of 9,428 metagenomes).”

Reference 16: "Pasolli, E. *et al.* Extensive Unexplored Human Microbiome Diversity Revealed by Over 150,000 Genomes from Metagenomes Spanning Age, Geography, and Lifestyle. *Cell* **176**, 649–662.e20 (2019)."

- on a related note, what is the evidence for a higher consumption of raw vegetables in the non-westernized cohorts? furthermore, are you certain that there was a high consumption of kimchi (a Korean dish) in these cohorts, as claimed in I. 144?

Response

We understand the Reviewer's point, thank you. We agree that in absence of dietary records for the subjects in the cohorts that we have included in this study, such statements may appear speculative. This is the reason why we gave it as a possibility by reporting "[...] their widespread in raw vegetables²³ and kimchi²⁴ that are likely consumed in such populations". In response to the Reviewer's concern, we have left out in this revised version the comment on kimchi, as it is a specific product whose consumption is not supported by dietary records. The sentence sounds as follows now:

(line 138) "which is consistent with their widespread prevalence in raw vegetables²³ that are likely consumed in such populations"

We would like to keep the comment on raw vegetables since non-western populations have agrarian diets or hunter-gatherer diet and lifestyle, which are recognized to be characterized by high consumption of tubers, drupes, roots and fruits (Broussard and Devkota, 2016 (<https://doi.org/10.1016/j.molmet.2016.07.007>); Schnorr et al., 2014 (<https://doi.org/10.1038/ncomms4654>)). It was also reported that non-Western African populations such as !Kung and Hadza still obtain 60-80% and 50-65% of their diet from plant foods, respectively (Crittenden and Schnorr, 2016 (<https://doi.org/10.1002/ajpa.23148>)). We added such references to support our statements in the revised version of the manuscript:

(line 140) "In fact, non-western populations usually have hunter-gatherer diet and lifestyle, which is recognized to be characterized by high consumption of tubers, drupes, roots, and fruits^{24,25}. Indeed, it was also reported that the !Kung and the Hadza, two non-Western African populations, still obtain 60–80% and 50–65% of their diet from plant foods, respectively²⁶."

Reference 24: "Broussard, J. L. & Devkota, S. The changing microbial landscape of Western society: Diet, dwellings and discordance. *Mol Metab* **5**, 737–742 (2016)."

Reference 25: "Schnorr, S. L. *et al.* Gut microbiome of the Hadza hunter-gatherers. *Nat. Commun.* **5**, 3654 (2014)."

Reference 26: "Crittenden, A. N. & Schnorr, S. L. Current views on hunter-gatherer nutrition and the evolution of the human diet. *Am. J. Phys. Anthropol.* **162**, 84–109 (2017)."

- I. 212: what does "most prevalent" refer to, i.e. where/in which SGB/foods was *B. linens* most prevalent? please clarify this sentence

Response

We modified the sentence as follows:

(line 210) "The most prevalent food-specific species was *Brevibacterium linens* (24 MAGs) which was reconstructed from multiple cheese types (i.e., surface ripened⁸, smear ripened¹⁴, hard, and tomme). Food-specific SGBs also included [...]"

Reference 8: "Bertuzzi, A. S. *et al.* Omics-Based Insights into Flavor Development and Microbial Succession within Surface-Ripened Cheese. *mSystems* **3**, (2018)."

Reference 14: "Wolfe, B. E., Button, J. E., Santarelli, M. & Dutton, R. J. Cheese rind communities provide tractable systems for in situ and in vitro studies of microbial diversity. *Cell* **158**, 422–433 (2014)."

- I. 332: what measure is the resemblance based on? nucleotide identity?

Response

Yes, we clarified this in the text:

(line 331): "Interestingly, the three MAGs of *Lc. garvieae* resembled more the strains found in non-westernized human populations in terms of nucleotide identity."

Figures / legends:

- figure 1: is the scale percentage?

Response

Yes the scale is in percentage, we added this information in **Figure 1** (we refer the Reviewer to the updated figure in the revised version of the manuscript).

- figure 2: I would prefer the scale of the bars in A and B to not be linear; especially in the bottom part of A, as well as B, it's not obvious from the axis label that it is - also it's kind of wrong to have the bars extend below 1. Potentially, the whole figure might be more appropriately presented as a table.

Response

We modified content and caption of **Figure 2** by clarifying as follows:

- X-axes for panels **A)** and **B)** are in logarithmic scale;
- Added the 0 to the axes. We feel the bars are less ambiguous now;
- Added the #human MAGs to panel A as suggested by Reviewer 1.

This is the updated figure:

Figure 2. Microbial genomes reconstructed from food metagenomes. A) Most prevalent species-level genome bins (SGBs) in 666 MAGs reconstructed from 303 food metagenomes and overlapping with human MAGs (i.e., found in at least one of the 154,723 human MAGs). **B)** Most prevalent food SGBs not overlapping with human MAGs. kSGBs denote SGBs with at least one reference microbial genome, whereas fSGBs identify newly assembled SGBs from food metagenomes only. X-axes for panels **A)** and **B)** are in logarithmic scale. **C)** Fraction of reference genomes per source type for the 30 selected LAB species and grouped by genera (the same plot at species-level is reported in **Figure S3**). Raw data in **Table S6** and **Table S7**.

- figure 3 and 4: please indicate in the figure legends what the cladograms are based on. If I understood it correctly, the trees are built based on marker gene distances and not whole-genome ANIs? this would be very important to mention.

Response

Yes, the phylogenetic trees shown in **Figure 3** and **Figure 4** were built using species-specific marker genes. We added this information in the captions:

Figure 3. Comparative genomic analysis of the two most prevalent LAB identified in the human gut microbiome. A) *S. thermophilus* is a genetically diverse species both in food and human sources with MAGs reconstructed from Asian gut metagenomes enriched in Clade A ($p < 1e-10$). **B)** *Lc. lactis* subsp. *lactis* is formed by three main clusters: Cluster 1 exhibits an overall low diversity and includes mostly food genomes related to cheese and dairy fermentation; Cluster 2 is dominated by environmental and raw vegetable products and more diverse human MAGs; Cluster 3 includes only two MAGs from nunu. Phylogenetic trees were built on species-specific marker genes and report five different metadata. Multidimensional scaling (MDS) on average nucleotide identity (ANI) distance is coloured with source information.

Figure 4. Comparative genomic analysis of relevant lactobacilli found in both food and human microbiomes. A) *Lb. delbrueckii* is not prevalent in the gut, and the only two

subspecies found in both food and human samples are subsps. *lactis* and *bulgaricus*. Subsp. *delbrueckii*, *sunkii*, and *jakobsenii* are found in food, but never reconstructed from the gut. **B)** *Lb. rhamnosus* exhibits the greatest number of genomes from human isolates but is scarcely reconstructed from metagenomes. A specific cluster identifies the LGG strain. **C)** *Lb. casei/paracasei* includes reference genomes identified as both *Lb. casei* and *Lb. paracasei*. We detect two main clusters both occurring in food and human samples. **D)** *Lb. helveticus* exhibits three main clusters, with Cluster 1 including all the dietary supplement strains (source in green), while food genomes are predominantly spread across the other two groups. Phylogenetic trees were built on species-specific marker genes and report five different metadata. Multidimensional scaling (MDS) on average nucleotide identity (ANI) distance is coloured with source information.

Language and structure:

- **abstract: I feel the abstract could do with more conclusion and less methodological data. I found it difficult to discern the main message other than "We analysed a lot of data" (which you did and that's great).**

Response

We would like to thank the reviewer for this comment. We have revised the abstract by leaving out some methodological details and numbers and by adding some more conclusive remarks:

“Lactic acid bacteria (LAB) are widely studied, they are fundamental in the production of fermented foods and several strains are regarded as probiotics. Large quantities of live LAB are consumed within fermented foods, but it is not yet known to what extent the LAB we ingest become members of the gut microbiome. By analysis of 9,445 metagenomes from human samples, we demonstrate that the prevalence and abundance of LAB species in stool samples is generally low and linked to age, lifestyle, and geography, with *Streptococcus thermophilus* and *Lactococcus lactis* being most prevalent. Moreover, we identify genome-based differences between food and gut microbes by considering 666 metagenome-assembled genomes (MAGs) newly reconstructed from fermented food microbiomes alongwith 154,723 human MAGs and 193,078 reference genomes. Our large-scale genome-wide analysis demonstrates that closely related LAB strains occur in both food and gut environments and provides unprecedented evidence that fermented foods can be indeed regarded as a possible source of LAB for the gut microbiome.”

- **I. 137: replace "resulted" with "were"**

Response

Thank you, we fixed this (see line 132)

- **I. 144 and I. 190: insert "occurrence" or "presence" behind "widespread"**

Response

Thank you, we added “presence” behind “widespread” in line 144 (line 139 now). We modified line 190 (line 187 now) as follows:

“Sixty MAGs were associated with *S. thermophilus* (N = 60); the majority of them was reconstructed from cheese and yogurt, and five additional genomes were extracted from different fermented foods such as wagashi, beetroot kvass, ryazhenka, ruž'a, and labne.”

- I. 172: remove superfluous "a"

Response

Thank you, we fixed this typo (see line 170).

- I. 185: insert "food" before "MAGs" for clarity

Response

Thank you, we added this (see line 183).

- I. 187: replace "mostly reconstructed" with "most often/commonly reconstructed"

Response

Thank you, we fixed this typo (see line 185).

- I. 190: replace "apart" with "despite"

Response

Thank you, we rephrased the sentence as replied previously.

- I. 205: replace "most reconstructed SGB" with "SGB with the most MAGs"

Response

Thank you, we fixed this (see line 203).

- I. 333: replace "scenario" with "situation"

Response

Thank you, we fixed this (see line 334).

- legend to figure 4: replace "included" in the last sentence with "including"

Response

Thank you, we fixed this (see line 615).

REVIEWERS' COMMENTS:

Reviewer #1 (Remarks to the Author):

The authors have addressed my concerns. They made the suggested changes to Figures 1, 2, 3 and 4, added appropriate p-values, included relevant data in the supplementary information, and fixed some minor corrections to the text. I thank the authors for their time.

Reviewer #2 (Remarks to the Author):

The authors addressed all of my few concerns.

Reviewer #3 (Remarks to the Author):

I thank you for thoroughly addressing all my comments. I have no further issues with the study and wish you a happy publication time.

REVIEWERS' COMMENTS:

Reviewer #1 (Remarks to the Author):

The authors have addressed my concerns. They made the suggested changes to Figures 1, 2, 3 and 4, added appropriate p-values, included relevant data in the supplementary information, and fixed some minor corrections to the text. I thank the authors for their time.

Reviewer #2 (Remarks to the Author):

The authors addressed all of my few concerns.

Reviewer #3 (Remarks to the Author):

I thank you for thoroughly addressing all my comments. I have no further issues with the study and wish you a happy publication time.

Response

We thank the Reviewers for the time devoted to revise our manuscript. We are happy to see that no further modifications or clarifications are required.